# CO3: Cooperative Unsupervised 3D Representation Learning for Autonomous Driving

**Runjian Chen** [1], **Yao Mu** [1], **Runsen Xu** [4], **Wenqi Shao** [2*], **Chenhan Jiang** [5],
**Hang Xu** [3], **Yu Qiao** [2], **Zhenguo Li** [3], **Ping Luo** [1*]

```
{rjchen, muyao}@connect.hku.hk   pluo.lhi@gmail.com
{shaowenqi, qiaoyu}@pjlab.org.cn
li.zhenguo@huawei.com   xbjxh@live.com
runsenxu@connect.cuhk.edu.hk   cjiangao@connect.hkust.hk
```

[1] The University of Hong Kong
[2] Shanghai AI Laboratory
[3] Huawei Noah's Ark Lab
[4] The Chinese University of Hong Kong
[5] Hong Kong University of Science and Technology

## Abstract

Unsupervised contrastive learning for indoor-scene point clouds has achieved great successes. However, unsupervised representation learning on outdoor-scene point clouds remains challenging because previous methods need to reconstruct the whole scene and capture partial views for the contrastive objective. This is infeasible in outdoor scenes with moving objects, obstacles, and sensors. In this paper, we propose CO3, namely **Co**operative **Co**ntrastive Learning and **Co**ntextual Shape Prediction, to learn 3D representation for outdoor-scene point clouds in an unsupervised manner. CO3 has several merits compared to existing methods. (1) It utilizes LiDAR point clouds from vehicle-side and infrastructure-side to build views that differ enough but meanwhile maintain common semantic information for contrastive learning, which are more appropriate than views built by previous methods. (2) Alongside the contrastive objective, we propose contextual shape prediction to bring more task-relevant information for unsupervised 3D point cloud representation learning and we also provide a theoretical analysis for this pre-training goal. (3) As compared to previous methods, representation learned by CO3 is able to be transferred to different outdoor scene datasets collected by different type of LiDAR sensors. (4) CO3 improves current state-of-the-art methods on both *Once*, *KITTI* and *NuScenes* datasets by up to 2.58 mAP in 3D object detection task and 3.54 mIoU in LiDAR semantic segmentation task. Codes and models will be released here. We believe CO3 will facilitate understanding LiDAR point clouds in outdoor scene.

## 1 Introduction

LiDAR is an important sensor for autonomous driving in outdoor environments and both of the machine learning and computer vision communities have shown strong interest on perception tasks on LiDAR point clouds, including 3D object detection, segmentation and tracking. Up to now, randomly initializing and directly *training from scratch* on detailed annotated data still dominates this field. On the contrary, recent research efforts (He et al., 2020; Tian et al., 2019; Caron et al., 2020; Grill et al., 2020; Wang et al., 2021) in image domain focus on unsupervised representation learning with contrastive objective on different views built from different augmentation of images. They pre-train the 2D backbone with a large-scale dataset like ImageNet (Deng et al., 2009) in an unsupervised manner and use the pre-trained backbone to initialize downstream neural networks on different datasets and tasks, which achieve significant performance improvement in 2D object detection and semantic segmentation (Girshick et al., 2014; Lin et al., 2017; Ren et al., 2015). Inspired by these

---

*corresponding authors are Wenqi Shao and Ping Luo

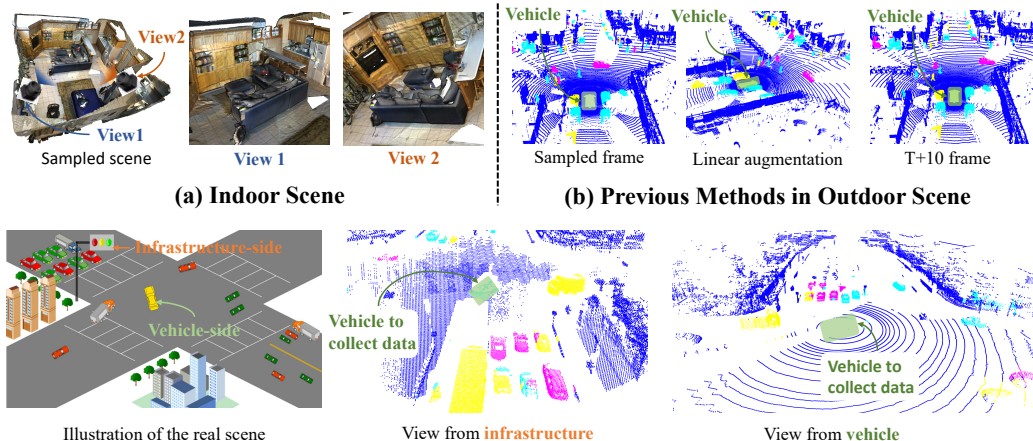

Figure 1: Example views built by different methods in contrastive learning, including (a) previous indoor-scene methods (b) previous outdoor-scene methods and (c) the proposed CO3. Compared to previous methods, CO3 can build two views that differ a lot and share adequate common semantics.

successes, we explore unsupervised representation learning for *outdoor scene* point clouds to learn general representations for different architectures on various downstream datasets and tasks.

In the past decade, learning 3D representation from unlabelled data has achieved great success in *indoor-scene* point clouds. PointContrast (Xie et al., 2020) is the pioneering work and proposes to reconstruct the whole indoor scenes, collect partial point clouds from two different poses and utilize them as two views in contrastive learning to learn **dense** (point-level or voxel-level) representation. More recent works such as (Hou et al., 2021) and (Liu et al., 2020) also need the reconstruction and this naturally assumes that the environment is static. Fig. 1 (a) shows an example of views in PointContrast (Xie et al., 2020). We can see that the views differ a lot because they are captured from different poses but meanwhile, they still contain enough common semantic information such as the same sofa and table. These are demonstrated important properties of views in contrastive learning in (Tian et al., 2020).

However, outdoor scenes are dynamic and large-scale, making it impossible to reconstruct the whole scenes for building views. Thus, methods in (Xie et al., 2020; Hou et al., 2021; Liu et al., 2020) cannot be directly transferred but there exists two possible alternatives to build views. The first idea, embraced by (Liang et al., 2021; Yin et al., 2022), is to apply data augmentation to single frame of point cloud and treat the original and augmented versions as different views, which are indicated by the first and second pictures in Fig. 1 (b). However, all the augmentation of point clouds, including random drop, rotation and scaling, can be implemented in a linear transformation and views constructed in this way do not differ enough. The second idea is to consider point clouds at different timestamps as different views, represented by (Huang et al., 2021). Yet the moving objects would make it hard to find correct correspondence for contrastive learning. See the first and third pictures in Fig. 1 (b), while the autonomous vehicle is waiting at the crossing, other cars and pedestrians are moving around. The autonomous vehicle has no idea about how they move and is not able to find correct correspondence (common semantics). Due to these limitations, it is still challenging when transferring the pre-trained 3D encoders to datasets collected by different LiDAR sensors. *Could we find better views to learn general representations for outdoor-scene LiDAR point clouds?*

In this paper, we propose **CO**operative **CO**ntrastive Learning and **CO**ntextual Shape Prediction, namely CO3, to explore the potential of utilizing vehicle-infrastructure cooperation dataset for building adequate views in unsupervised 3D representation learning. As shown in (c) in Fig. 1, a recently released infrastructure-vehicle-cooperation dataset called DAIR-V2X (Yu et al., 2022) is utilized to learn general 3D representations. Point clouds from both vehicle and infrastructure sides are captured at the same timestamp thus views share adequate common semantic. Meanwhile infrastructure-side and vehicle-side point clouds differ a lot. These properties make views constructed in this way appropriate in contrastive learning. Besides, as proposed in (Wang et al., 2022), representations learned by pure contrastive learning lack task-relevant information. Thus we further add a pre-training goal

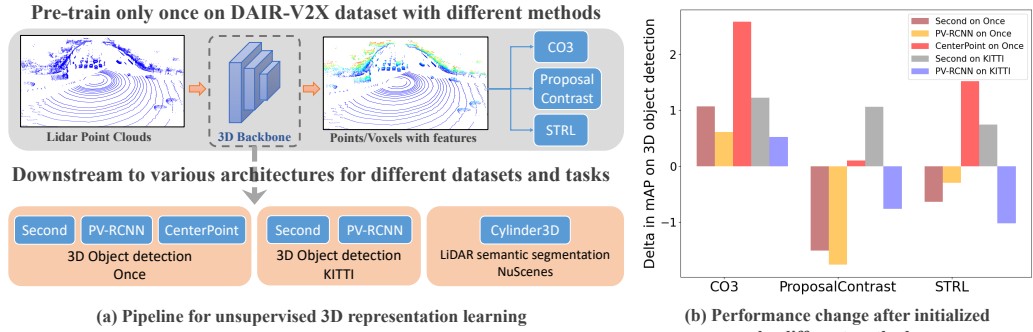

Figure 2: (a) shows the unsupervised 3D representation learning pipeline. (b) presents the performance changes after pre-training with different methods. Our CO3 achieves consistent improvement.

called contextual shape prediction to reconstruct local point distribution, encouraging our proposed CO3 to capture more task-relevant information.

Our contributions can be summarized as follows. (1) CO3 is proposed to utilize the vehicle-infrastructure-cooperation dataset to build adequate views for unsupervised contrastive 3D representation learning on *outdoor-scene* point clouds. (2) A shape-context prediction task is proposed alongside to inject task-relevant information, which is beneficial for downstream 3D detection and LiDAR semantic segmentation tasks. (3) The learned 3D representations is generic enough to be well transferred to datasets collected by different LiDAR sensors. (4) Extensive experiments demonstrate the effectiveness of CO3. For example, on 3D object detection task, CO3 improves Second, Center-Points on *Once* dataset by 1.07 and 2.58 mAPs respectively. As for LiDAR semantic segmentation task, CO3 improves Cylinder3D on *NuScenes* dataset by 3.54 mIoUs.

## 2 RELATED WORKS

**3D Perception Tasks.** *3D object detection* aims to predict 3D boundary boxes for different objects in the LiDAR point clouds. Current 3D detectors can be divided into three main streams due to the 3D backbones they use: (1) point-based methods (Shi et al., 2019; Chen et al., 2017; Yang et al., 2018) use point-based 3D backbone. (2) voxel-based methods (Zhou & Tuzel, 2018; Lang et al., 2019; Su et al., 2015; Shi et al., 2020b; Yin et al., 2021; Fan et al., 2021) generally transform point cloud into voxel grids and process them using 3D volumetric convolutions. (3) point-voxel-combined methods (Shi et al., 2020a; 2021; Deng et al., 2021) utilize features from both (1) and (2). *LiDAR Semantic Segmentation* aims to predict per-point label for LiDAR point clouds. Cylinder3D (Zhu et al., 2021) and PVKD (Hou et al., 2022) are current SOTA methods for LiDAR Semantic Segmentation.

**Unsupervised 3D Representation Learning.** As shown in (a) of Fig. 2, unsupervised 3D representation learning aims to pre-train only for one time and downstream to different architectures on various datasets and tasks to achieve performance gain. PointContrast (Xie et al., 2020) is the pioneering work for unsupervised contrastive learning on *indoor-scene* point clouds, which relies on the reconstructed point clouds for constructing adequate views. To extend their ideas to outdoor-scene point clouds, GCC-3D and ProposalContrast (Liang et al., 2021; Yin et al., 2022) augment single frame of point cloud to build views and STRL (Huang et al., 2021) utilizes point clouds at different timestamps as views for contrastive learning. Alongside contrastive learning, authors in (Hu et al., 2021) propose to use safe space prediction as a pretext task for self-supervised representation learning. However, as discussed in Sec. 1, previous works fail to build suitable views and their learned representations are unable to transfer to datasets collected by different LiDAR sensors. In this paper, we propose to use point clouds from vehicle and infrastructure to construct views in contrastive learning and also extend the idea of shape context to introduce task-relevant information with a local-distribution prediction goal. The performance change after initialized with different methods are shown in (b) of Fig. 2. CO3 generally improve the performance and achieve more performance gains than other pre-training methods for different detectors on different datasets.

## 3 METHODS

In this section, we introduce the proposed CO3 for unsupervised representation learning on LiDAR point clouds in outdoor scenes. As detailed in Fig. 3, CO3 has two pre-training objectives: (a) a

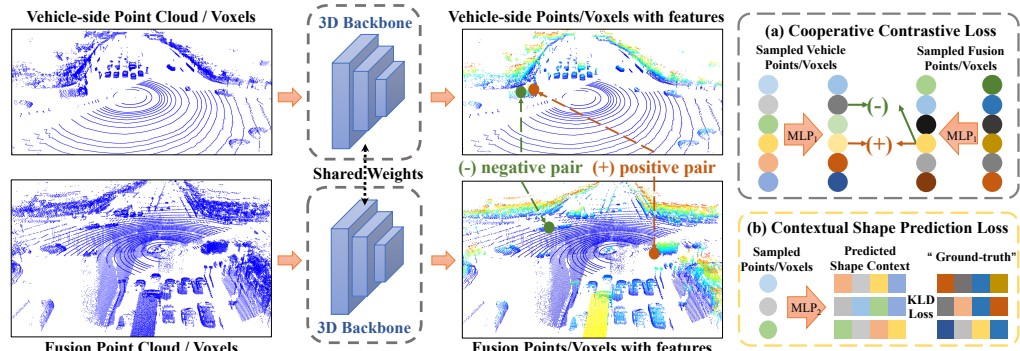

Figure 3: The pipeline of CO3. With vehicle-side and fusion point clouds as inputs, we first process them with the 3D backbone and propose two pre-training objectives: (a) Cooperative Contrastive Loss (b) Contextual Shape Prediction Loss

cooperative contrastive learning goal on dense (point-level or voxel-level) representations between vehicle-side and fusion point clouds, which provides adequate views for contrastive learning. (b) a contextual shape prediction loss to bring in more task-relevant information. To start with, we discuss the problem formulation and overall pipeline in Sec. 3.1. Then we respectively introduce the cooperative contrastive objective and contextual shape prediction goal in Sec. 3.2 and Sec. 3.3.

## 3.1 PROBLEM FORMULATION AND PIPELINE

**Notation.** To begin with, we denote LiDAR point clouds $\mathbf{P} \in \mathbb{R}^{N \times (3+d)}$ as the concatenation of the $xyz$-coordinate $\mathbf{C} \in \mathbb{R}^{N \times 3}$ and the point features $\mathbf{F} \in \mathbb{R}^{N \times d}$, which leads to $\mathbf{P} = [\mathbf{C}, \mathbf{F}]$. Here $N$ denotes the number of points (or voxels) and $d$ is the number of point feature channels. For raw LiDAR point clouds, $d = 1$ represents the intensity of each point. Moreover, we use the subscripts (e.g. 'v' and 'i') to indicate point clouds from different sources. For example, $\mathbf{P}_v = [\mathbf{C}_v, \mathbf{F}_v] \in \mathbb{R}^{N_v \times (3+d)}$ and $\mathbf{P}_i = [\mathbf{C}_i, \mathbf{F}_i] \in \mathbb{R}^{N_i \times (3+d)}$ respectively represent $N_v$ and $N_i$ points (or voxels) in vehicle and infrastructure sides. Besides, each pair of vehicle-side and infrastructure-side point clouds is associated with a transformation $\mathcal{T}$ mapping infrastructure-side coordinate to that of vehicle-side.

**Pre-processing and Encoding.** For cooperative learning, we need to first align the point clouds from both sides in the same coordinate. Hence, we transform the infrastructure-side point clouds to $\mathbf{P}'_i = [\mathbf{C}'_i, \mathbf{F}_i]$ where $\mathbf{C}'_i = \mathcal{T}(\mathbf{C}_i)$. However, we empirically find that the contrastive learning built upon vehicle-side point cloud $\mathbf{P}_v$ and transformed infrastructure-side point clouds $\mathbf{P}'_i$ only achieve marginal performance gains than training from scratch on downstream ONCE dataset (0.53 mAPs vs 2.58 mAPs, also see Table 7 in Appendix D). We believe this stems from the sparsity of LiDAR point clouds, which sometimes make it difficult to find good positive pairs to perform contrastive learning. To mitigate this problem, we concatenate the LiDAR point clouds from both sides to fusion point clouds $\mathbf{P}_f = [\mathbf{P}_v, \mathbf{P}'_i] \in \mathbb{R}^{(N_v + N_i) \times (3+d)}$, which is used as the contrastive view of vehicle-side point clouds $\mathbf{P}_v$ as shown in Fig.3. By default of notations, we can express the coordinate and feature of fusion point clouds as $\mathbf{C}_f = [\mathbf{C}_v, \mathbf{C}'_i]$ and $\mathbf{F}_f = [\mathbf{F}_v, \mathbf{F}_i]$, respectively. Then $\mathbf{P}_v$ and $\mathbf{P}_f$ are embedded by the 3D encoder $f^{enc}$ to obtain their 3D representations. We use subscript 'v/f' to indicate that the same operation is applied respectively on both pointclouds.

$$\hat{\mathbf{P}}_{v/f} = f^{enc}(\mathbf{P}_{v/f}) \tag{1}$$

where $\hat{\mathbf{P}}_{v/f} \in \mathbb{R}^{\hat{N}_{v/f} \times (3+\hat{d})}$ indicates the vehicle point cloud and fusion point cloud respectively. Here we use $\hat{N}_{v/f}$ to denote the number of points after encoding because pooling operation often exists in 3D encoders (Graham et al., 2018) and changes the number of voxels/points. Moreover, $\hat{d}$ is the number of feature channels after encoding.

**Loss Function.** To guide the 3D encoder to learn good representations in an unsupervised manner, our proposed CO3 consists of a cooperative contrastive loss $\mathcal{L}_{CO_2}$ and a contextual shape prediction loss $\mathcal{L}_{CSP}$. The overall loss function is given by:

$$\mathcal{L} = \frac{1}{|\mathcal{P}_{v/f}|} \sum_{\mathbf{P}_{v/f} \in \{\mathcal{P}_{v/f}\}} \mathcal{L}_{CO_2}\{\hat{\mathbf{P}}_v, \hat{\mathbf{P}}_f\} + w \times \mathcal{L}_{CSP}\{\hat{\mathbf{P}}_v, \hat{\mathbf{P}}_f, \mathbf{C}_f\} \tag{2}$$

where $\mathcal{P}_{\text{v/f}}$ denote a batch of vehicle and fusion point clouds and $|\mathcal{P}_{\text{v/f}}|$ indicates the batch size. $\mathcal{L}_{\text{CO}_2}$ applies contrastive learning on the encoded vehicle and fusion pointclouds. Meanwhile, $\mathcal{L}_{\text{CSP}}$ introduces more task-relevant information into $f^{\text{enc}}$ by using the encoded features to predict contextual shape obtained by the coordinate of fusion point clouds $\mathbf{C}_{\text{f}}$. $w$ is a weight to balance the losses.

## 3.2 COOPERATIVE CONTRASTIVE OBJECTIVE

Unsupervised contrastive learning has been demonstrated successful in image domain (He et al., 2020; Tian et al., 2019) and indoor-scene point clouds (Xie et al., 2020). However, when it turns to outdoor-scene LiDAR point clouds, building adequate views, which share common semantics while differing enough, for contrastive learning is difficult. To tackle this challenge, we utilize a recently released vehicle-infrastructure-cooperation dataset called DAIR-V2X (Yu et al., 2022) and use vehicle-side point clouds and fusion point clouds as views for contrastive representation learning. Views built in this way differ a lot because they are captured at different positions and they share enough information because they are captured at the same timestamp. More details about view building in contrastive learning can be found in Appendix A.

**Contrastive Head.** Following BYOL (Grill et al., 2020), we construct contrastive head of cooperative contrastive objective by a Multi-Layer-Perceptron (MLP) layer and a $\ell_2$-normalization. Specifically, the embedded features of vehicle and fusion point clouds, $\hat{\mathbf{F}}_{\text{v}}$ and $\hat{\mathbf{F}}_{\text{f}}$, are first projected by a MLP layer denoted as $\text{MLP}_1$. Then an $\ell_2$-normalization is applied along feature dimension. This process is described below, where $\mathbf{Z}_{\text{v/f}} \in \mathbb{R}^{\hat{N}_{\text{v/f}} \times d_z}$ and $d_z$ is the output feature dimension.

$$\hat{\mathbf{Z}}_{\text{v/f}} = \text{MLP}_1(\hat{\mathbf{F}}_{\text{v/f}}), \quad \mathbf{Z}_{\text{v/f}} = \hat{\mathbf{Z}}_{\text{v/f}}/\|\hat{\mathbf{Z}}_{\text{v/f}}\|_2 \qquad (3)$$

**Cooperative Contrastive Loss.** For cooperative contrastive learning, we obtain positive pairs by exploring the correspondence between coordinates $\hat{\mathbf{C}}_{\text{v}}$ and $\hat{\mathbf{C}}_{\text{f}}$. In detail, we uniformly sample $N_1$ points from the vehicle point cloud and find their corresponding points in the fusion point cloud to form $N_1$ pairs of features $(\{z_{\text{v/f}}^n\}_{n=1}^{N_1})$ from $\mathbf{Z}_{\text{v/f}}$ for contrastive learning, where $z_{\text{v/f}}^n \in \mathbb{R}^{1 \times d_z}$. We treat corresponding points (or voxels) as positive pairs and otherwise negative pairs for contrastive learning. Example pairs are shown in Fig. 3. The final loss function is formulated below,

$$\mathcal{L}_{\text{CO}_2} = \frac{1}{N_1} \sum_{n=1}^{N_1} -\log\left(\frac{\exp(z_{\text{v}}^n \cdot z_{\text{f}}^n/\tau)}{\sum_{i=1}^{N_1} \exp(z_{\text{v}}^n \cdot z_{\text{f}}^i/\tau)}\right) \qquad (4)$$

where $\tau$ is the temperature parameter. The bottom right box in Fig. 3 shows examples for this loss. As ground points only contain background information that is not related to perception tasks, we mark those points with height value lower than a threshold $z_{\text{thd}}$ as ground points and filter them out when sampling.

## 3.3 CONTEXTUAL SHAPE PREDICTION

CO3 aims to learn representations applicable to various downstream datasets. However, as demonstrated in (Wang et al., 2022), pure contrastive loss in Eqn. (4) would result in the lack of task-relevant information in the learned representations, making it hard to generalize across different architectures and datasets. Meanwhile, an additional reconstruction objective could increase the mutual information between the representations and the input views, which would bring in task-relevant information. Please refer to detailed explanations about this in Appendix B. For outdoor-scene point clouds, it is difficult to reconstruct the whole scene with point/voxel-level representations. To mitigate this issue, we propose to reconstruct the neighborhood of each point/voxel with its representation.

**Local Distribution.** Shape context has been demonstrated as useful point feature descriptor in previous works (Hou et al., 2021; Belongie et al., 2002; Körtgen et al., 2003; Xie et al., 2018). Fig. 4 shows two examples of shape context with 8 bins (the number of bins can be changed) around the query point, which

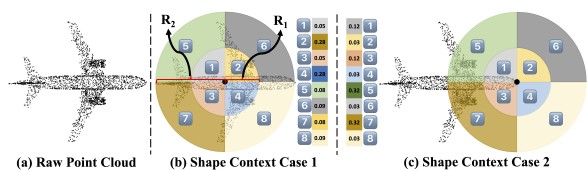

Figure 4: Two examples of shape context.

is marked as a larger black point. Previous works use the number of points in each bin as the descriptor for the query point. However, it can be difficult for the neural net to learn to regress the exact number of points in each bin. Thus, we propose to use the representation to predict a local distribution built upon shape context. In practice, we divide the neighborhood of each point in fusion point clouds into $N_{\text{bin}} = 32$ bins along $xyz$-plane with $R_1 = 0.5m$ and $R_2 = 4m$. Then we compute the "raw" shape context $Q' \in \mathbb{R}^{N_f \times N_{\text{bin}}}$, which describe the number of points in each bin around every fusion point. Next, we apply $\ell_2$-normalization to $Q'$ and a consecutive softmax to obtain the ground-truth shape context $Q \in \mathbb{R}^{N_f \times N_{\text{bin}}}$ as described below, where $Q_{i,*} \in \mathbb{R}^{1 \times N_{\text{bin}}}$ describes the ground-truth local distribution for $i$-th point.

$$Q_{i,*} = \text{softmax}(Q'_{i,*}/\|Q'_{i,*}\|_2) \tag{5}$$

**Prediction Loss.** We use the encoded features of both vehicle and fusion point clouds, i.e. $\hat{\mathbf{F}}_{\text{v/f}}$, to predict the local distribution. To be specific, $\hat{\mathbf{F}}_{\text{v/f}}$ is first passed through a MLP layer $\text{MLP}_2$ and softmax operation is applied on the projected features to get the predicted local distribution.

$$P_{\text{v/f}} = \text{softmax}(\text{MLP}_2(\hat{\mathbf{F}}_{\text{v/f}})) \tag{6}$$

where $P_{\text{v/f}} \in \mathbb{R}^{\hat{N}_{\text{v/f}} \times N_{\text{bin}}}$. Then we uniformly sample $N_2$ predictions ($\{p_{\text{v/f}}^n\}_{n=1}^{N_2}$) from $P_{\text{v/f}}$ and find their corresponding "ground truth" in $Q$ by coordinate correspondence, where we have $\{q_{\text{v/f}}^n\}_{n=1}^{N_2}$. The dimensions of each prediction and "ground truth" are $p_{\text{v/f}}^n \in \mathbb{R}^{1 \times N_{bin}}$ and $q_{\text{v/f}}^n \in \mathbb{R}^{1 \times N_{bin}}$. Finally, the prediction loss with KL-divergence is given by,

$$\mathcal{L}_{\text{CSP}} = \frac{1}{N_2} \sum_{n=1}^{N_2} \sum_{m=1}^{N_{\text{bin}}} (p_{\text{v}}^{n,m} \log \frac{p_{\text{v}}^{n,m}}{q_{\text{v}}^{n,m}} + p_{\text{f}}^{n,m} \log \frac{p_{\text{f}}^{n,m}}{q_{\text{f}}^{n,m}}) \tag{7}$$

## 4 EXPERIMENTS

The goal for unsupervised representation learning is to learn general representation that can benefit different downstream architectures on different downstream datasets and tasks. In this section, we design experiments to answer the question whether CO3 learns such representation as compared to previous methods. We first provide experiment setups in Sec. 4.1 and then discuss main results in Sec. 4.2. We also conduct ablation study and qualitative visualization in Sec. 4.3 and 4.4.

### 4.1 EXPERIMENT SETUP

**Pre-training Dataset.** We utilize the recently released vehicle-infrastructure-cooperation dataset called *DAIR-V2X* (Yu et al., 2022) to pre-train the 3D encoder. *DAIR-V2X* is the first real-world autonomous dataset for vehicle-infrastructure-cooperative task. The LiDAR sensor at vehicle-side is 40-beam while a 120-beam LiDAR is utilized at infrastructure-side. There are 38845 LiDAR frames (10084 in vehicle-side and 22325 in infrastructure-side) for cooperative-detection task. The dataset contains around 7000 synchronized cooperative samples in total.

**Implementation Details of CO3.** We use Sparse-Convolution as the 3D encoder which is a 3D convolutional network because it is widely used as 3D encoders in current state-of-the-art methods (Zhou & Tuzel, 2018; Yin et al., 2021; Shi et al., 2020a). We set the number of feature channels $d^{enc} = 64$, the temperature parameter in contrastive learning $\tau = 0.07$, the dimension of common feature space of vehicle-side and fusion point clouds $d_z = 256$ and the sample number in cooperative contrastive loss $N_1 = 2048$. For contextual shape prediction, we set the number of bins $N_{bin} = 32$, the sample number $N_2 = 2048$ and the weighting constant $w = 10$. The threshold for ground point filtering is $z_{\text{thd}} = 1.6m$. We empirically find that freezing the parameters of $\text{MLP}_2$ brings better results in detection task thus we fix them.

**Baselines.** We implement STRL (Huang et al., 2021) and use the official code of ProposalContrast (Yin et al., 2022) as baselines. Besides, as proposed in (Mao et al., 2021), several methods in image domain and indoor-scene point clouds can be transferred to outdoor scene point clouds, including Swav (Caron et al., 2020), Deep Cluster (Caron et al., 2018), BYOL (Grill et al., 2020) and Point Contrast (Xie et al., 2020). ***Note*** that in order to make fair comparisons, all the pre-training methods are pre-trained on *DAIR-V2X*. Thus there might exist number discrepancy between the results and previous benchmarks.

| Init. | Second | | | PV-RCNN | | | CenterPoint | | |
|---|---|---|---|---|---|---|---|---|---|
| | **Overall** | 0-30m | 30-50m | **Overall** | 0-30m | 30-50m | **Overall** | 0-30m | 30-50m |
| Rand | 52.21 | 60.71 | 47.31 | 54.55 | 63.05 | 50.00 | 55.92 | 66.39 | 50.16 |
| Swav | 53.03 | 61.38 | 48.34 | 54.89 | 63.41 | 50.31 | 57.00 | 67.54 | 50.60 |
| Deep Cluster | 52.30 | 61.03 | 47.00 | 54.91 | 63.84 | 50.29 | 57.65 | 68.05 | 51.04 |
| BYOL | 45.24 | 53.84 | 40.22 | 49.41 | 58.04 | 45.19 | 52.17 | 63.53 | 45.00 |
| PointContrast | 47.64 | 56.91 | 42.54 | 50.49 | 58.94 | 46.30 | 54.17 | 65.57 | 46.53 |
| ProposalContrast | 50.70 | 59.42 | 46.09 | 52.79 | 60.97 | 48.91 | 56.02 | 66.26 | 48.70 |
| STRL | 51.57 | 59.80 | 46.59 | 54.25 | 63.03 | 49.31 | 57.44 | 67.90 | 50.47 |
| Ours | **53.28**$^{+1.07}$ | 62.16 | 49.33 | **55.17**$^{+0.61}$ | 63.94 | 50.29 | **58.50**$^{+2.58}$ | 69.09 | 51.51 |

Table 1: Results of 3D object detection on Once dataset (Mao et al., 2021). We conduct experiments on 3 different detectors: Second (Zhou & Tuzel, 2018) (short as Sec.), PV-RCNN (Shi et al., 2021) (short as PV) and CenterPoint (Yin et al., 2021) (short as Cen.) and 8 different initialization methods including random (short as Rand, i.e. training from scratch), Swav (Caron et al., 2020), Deep Cluster (short as D. Cl.) (Caron et al., 2018), BYOL (Grill et al., 2020), Point Contrast (short as P.C.) (Xie et al., 2020), GCC-3D (Liang et al., 2021) and STRL (Huang et al., 2021). Results are mAPs in %. "0-30m" and "30-50m" respectively indicate results for objects in 0 to 30 meters and 30 to 50 meters. The "Overall" metric highlighted in red is the overall mAP, which serves as major metric for comparisons. We use bold font for the best overall mAP of each detector for better understanding.

**Downstream Tasks.** Two downstream tasks are selected for evaluation: 3D object detection and LiDAR semantic segmentation. 3D object detection task takes raw 3D point clouds as inputs and aims to output 3D boundary boxes of different objects in the scene. LiDAR semantic segmentation assign each 3D point a category label, including Car, Pedestrian, Bicycle, Truck and so on.

**3D Object Detection.** We select two downstream datasets: Once (Mao et al., 2021) and KITTI (Geiger et al., 2012). *Once* has 15k fully annotated frames of LiDAR scans. A 40-beam LiDAR is used to collect the point cloud data. We adopt common practice, including point cloud range and voxel size, in their public code repository. mAPs (mean accurate precisions) in different ranges and overall mAP are presented. *KITTI* is a widely used self-driving dataset, where point clouds are collected by LiDAR with 64 beams. It contains around 7k samples for training and another 7k for evaluation. All the results are evaluated by mAPs with three difficulty levels: Easy, Moderate and Hard. We select several current state-of-the-art methods implemented in the public repository of Once dataset (Mao et al., 2021)[1] and OpenPCDet[2] to evaluate the quality of representations learned by CO3, including Second (Zhou & Tuzel, 2018), CenterPoint (Yin et al., 2021) and PV-RCNN (Shi et al., 2020a).

**LiDAR Semantic Segmentation.** We select NuScenes (Caesar et al., 2020) as downstream dataset. There are 1000 scenes each of which lasts 20 seconds in *NuScenes*. It is collected with a 32-beam LiDAR sensor and the total number of frames is 40,000. We select Cylinder3D (Zhu et al., 2021) as downstream architecture. The full training is time-consuming (at least 4 days for one training) and we use a **1/8 training schedule** setting to evaluate whether CO3 is able to speed up training. We follow the conventional evaluation metrics. mAPs for detailed categories and mean intersection-over-union (mIoU) for overall evaluation are presented. Per-class mIoU is first computed as $mIoU_i = \frac{TP_i}{TP_i + FP_i + FN_i}$, where $TP_i$, $FP_i$ and $FN_i$ respectively represent true positive, false positive and false negative for class $i$. We then average over classes and get the final mIoU.

## 4.2 MAIN RESULTS

**Once Detection.** As shown in Table 1, when initialized by CO3 , all the three detectors achieve the best performance on the overall mAPs, which we value the most, and CenterPoint (Yin et al., 2021) achieves the highest overall mAP (58.50) with 2.58 improvement. The improvement on PV-RCNN (Shi et al., 2021) is 0.62 in mAP (similar lower improvement with other pre-training methods) because PV-RCNN (Shi et al., 2021) has both the point-based and voxel-based 3D backbones, among which CO3 only pre-trains the voxel-based branch. It can be found that other baselines are not able to learn general representation for different architectures. For example, STRL achieves +1.52 mAPs improvements on CenterPoint but degrades the performance of Second and PV-RCNN. On the contrary, CO3 achieve consistent improvement over different detectors.

**KITTI Detection.** As shown in Table 2, when initialized by CO3 , PV-RCNN (Shi et al., 2021) achieves the best performance on Easy and Hard (+1.19) level and third place on Moderate level.

---

[1] https://github.com/PointsCoder/Once_Benchmark
[2] https://github.com/open-mmlab/OpenPCDet

| Initialization | Second | | | PV-RCNN | | |
|---|---|---|---|---|---|---|
| | Easy | Moderate | Hard | Easy | Moderate | Hard |
| Random | 73.29 | 63.16 | 60.34 | 78.54 | 67.23 | 63.68 |
| Swav | 73.23 | 64.05 | **60.90** | 78.43 | **67.91** | 64.60 |
| Deep Cluster | 73.19 | 63.37 | 60.08 | 77.05 | 67.06 | 64.50 |
| BYOL | 71.05 | 60.39 | 56.98 | 77.96 | 67.50 | 64.42 |
| PointContrast | 72.67 | 62.74 | 59.21 | 77.62 | 67.79 | 63.31 |
| ProposalContrast | 74.23 | 64.22 | 60.88 | 77.28 | 66.47 | 63.22 |
| STRL | 73.95 | 63.90 | **60.90** | 77.10 | 66.21 | 62.90 |
| Ours | **74.40**$^{+1.11}$ | **64.38**$^{+1.22}$ | **60.90**$^{+0.56}$ | **78.84**$^{+0.30}$ | 67.75$^{+0.52}$ | **64.87**$^{+1.09}$ |

Table 2: Results of 3D object detection on KITTI dataset (Geiger et al., 2012). Results are mAPs in %. "Easy", "Moderate" and "Hard" respectively indicate difficulty levels defined in KITTI dataset. We use bold font for the best mAP of each detector in each difficulty level for better understanding.

| Initialization | **mIoU** | Car | Truck | Con. Veh. | Ped. | Trailer | Bic. | M.C. | S.W. | Terrain | Veg. |
|---|---|---|---|---|---|---|---|---|---|---|---|
| Random | 63.34 | 84.32 | 70.25 | 28.29 | 64.46 | 41.35 | 0.00 | 60.46 | 69.76 | 71.21 | 83.32 |
| P.C. | 64.31 | 84.70 | 74.36 | 30.81 | 63.52 | 47.13 | 10.16 | 55.55 | 70.38 | 71.43 | 84.65 |
| STRL | 64.71 | 84.66 | 76.65 | 27.30 | 63.29 | 52.76 | 12.79 | 60.11 | 70.27 | 71.70 | 84.60 |
| Ours | **66.88**$^{+3.54}$ | **85.52** | **77.00** | **36.00** | **66.93** | **53.13** | **19.51** | **70.65** | **70.40** | **72.43** | **84.88** |

Table 3: LiDAR semantic segmentation on NuScenes (Caesar et al., 2020). Results are mIoU and mAPs in %. We shows details results in each category. "Con. Veh.", "Ped.", "Bic.", "M.C.", "S. W." and "Veg." are abbreviations respectively for Construction Vehicle, Pedestrian, Bicycle, MotoCycle, Sidewalk and Vegetation. We use bold font for the best results in each column for better understanding.

| Init. | **Overall** | Vehicle | Pedestrian | Cyclist |
|---|---|---|---|---|
| Random | 55.92 | 62.85 | 45.52 | 59.39 |
| Sup. | 57.50 | 63.86 | 46.96 | 61.68 |
| CO3 | **58.50** | **64.60** | **48.83** | **62.17** |

Table 4: Comparison to supervised initialization

| Init. | **Overall** | Vehicle | Pedestrian | Cyclist |
|---|---|---|---|---|
| From Scratch | 55.92 | 62.85 | 45.52 | 59.39 |
| CO3 w. ground | 57.37 | 63.19 | 48.16 | 60.76 |
| CO3 | **58.50** | **64.60** | **48.83** | **62.17** |

Table 5: Ablation study on filtering out ground points.

Meanwhile, when Second (Zhou & Tuzel, 2018) is equipped with CO3, it achieves the highest mAPs on Easy level (+1.11), Moderate level (+1.22) and Hard level (+0.56). The lower gains on the KITTI dataset (Geiger et al., 2012) stem from the smaller number of training samples (half of that in Once (Mao et al., 2021)), which makes the detectors easily reach their capacity and improvement is hard to achieve. Consistent results across different initialization methods demonstrate this.

**NuScene Semantic Segmentation.** As shown in Table 3, CO3 improve Cylinder3D by 3.54 in mIoU and also achieves the best performance among the four initialization methods. Meanwhile, when initialized by CO3, Cylinder3D achieves the best mAPs among all the categories. On truck and construction vehicle, CO3 improve the performance of random initialization by 6.75 and 7.71 mAPs, which is very important in autonomous driving because correct segmentation of different vehicles can help benefit control and avoid accidents.

**Comparison to Supervised Pre-training.** We find backbone pre-trained on 3D detection task from the official codebase of DAIR-V2X dataset and use it to initialize CenterPoint, after which we fine-tune it on Once dataset. Results are shown in Table 4. It can be found that although supervised pre-training achieve improvement as compared to training from scratch, CO3 leads to the best results in all categories and the overall mAP. This is because supervised pre-training overfit to DAIR-V2X dataset, making the improvement lower than CO3 in 3D detection task on Once dataset.

**Overall Evaluation.** To summarize, CO3 achieves consistent performance gains over different architectures on different tasks (3D object detection and LiDAR semantic segmentation) and datasets (Once, KITTI and NuScenes) when pre-trained only on DAIR-V2X dataset. In comparison, other baseline pre-training methods only occasionally improve the performance and sometimes even bring degradation. These demonstrate CO3 learns general 3D representations.

## 4.3 ABLATION STUDY

**The influence of ground points in pre-training.** We first conduct ablation experiments on filtering out ground points. We use CO3 to pre-train 3D backbone without filtering out ground points and downstream it to 3D object detection with CenterPoint on Once. Results are shown in Table 5. When pre-training without filtering out ground points, the performance of CO3 drop in each category and

| Init. | Once (CenterPoint) | | | | KITTI (Second) | | | |
|---|---|---|---|---|---|---|---|---|
| | **Overall** | Vehicle | Pedestrian | Cyclist | **Overall** | Vehicle | Pedestrian | Cyclist |
| Random | 55.92 | 62.85 | 45.52 | 59.39 | 63.16 | 77.45 | 48.71 | 63.32 |
| Contextual Shape Prediction Only | 57.30 | 62.86 | **49.17** | 59.86 | 63.36 | 77.75 | 49.16 | 63.18 |
| Cooperative Contrastive Only | 57.53 | 63.39 | 48.14 | 61.05 | 63.41 | 77.40 | 47.78 | 65.06 |
| CO3 | **58.50** | **64.50** | 48.83 | **62.17** | **64.38** | **77.95** | **49.59** | **65.60** |

Table 6: Results of ablation study on Once (Mao et al., 2021) and KITTI (Geiger et al., 2012). We use CenterPoint (Yin et al., 2021) on Once and Second (Zhou & Tuzel, 2018) on KITTI. Results are mAPs in %. For Once, results are average across different ranges. For KITTI, results are all in moderate level. We highlight the best performance in each column for better understanding.

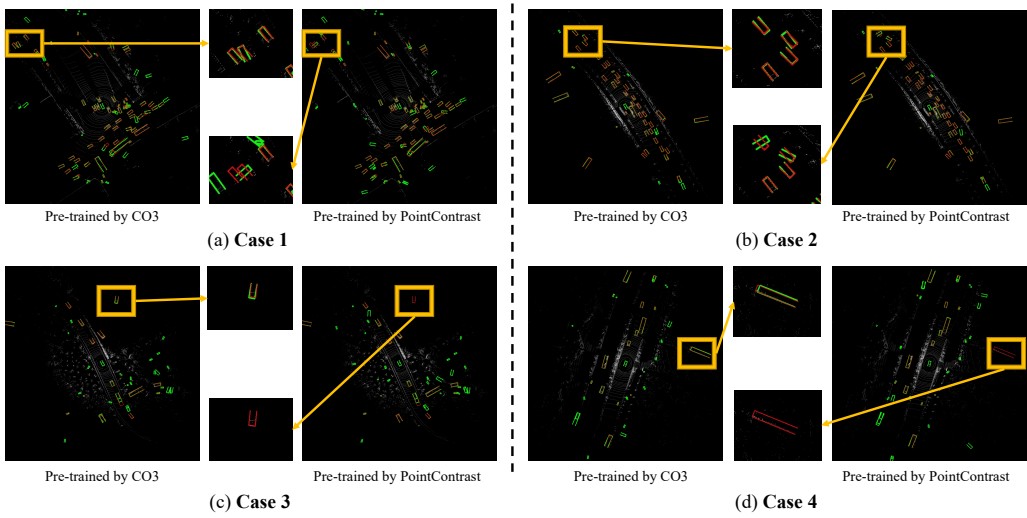

Figure 5: Visualization for detection results. Green boxes are predicted ones and red boxes are the ground truth.

the overall mAP. This demonstrate the effectiveness of filtering out ground points because ground points contain background information that is not useful for 3D perception tasks.

**The effect of each component in CO3.** We conduct ablation experiments to analyze the effectiveness of different components. We respectively pre-train the 3D encoder with cooperative contrastive objective and contextual shape prediction objective. As shown in Table 6, it can be found that each objective alone can achieve slight improvement, which demonstrates the effectiveness of either part. Besides, when pre-trained by CO3, we achieve the best performance on the overall mAPs. A more detailed discussion on the ablation study is provided in Appendix G

## 4.4 QUALITATIVE EXPERIMENT

We use the 3D backbone pre-trained by CO3 and PointContrast to initialize CenterPoint and train it on Once dataset. Then we visualize the detection results in Fig. 5, where predicted boxes are marked as green and the ground truth boxes are red. In Case 1 and 2, when the detector is initialized by CO3, the detection results are more correct in headings as shown in the zoom-in area. Correct heading prediction is important especially for control in autonomous driving. In Case 3 and 4, it can be found that CO3 helps the CenterPoint detect object with only a few points captured by the LiDAR sensor and meanwhile, PointContrast initialization fails to detect them. This is also essential in autonomous driving because detection failure can sometimes lead to disaster.

## 5 CONCLUSION AND FUTURE WORK

In this paper, we propose CO3, namely **Co**operative **Co**ntrastive Learning and **Co**ntextual Shape Prediction, for unsupervised 3D representation learning in outdoor scenes. The recently released vehicle-infrastructure-cooperation dataset DAIR-V2X is utilized to build views for cooperative contrastive learning. Meanwhile the contextual shape prediction objective provides task-relevant information for the 3D encoders. Our experiments demonstrate that the representation learned by CO3 can be transferred to various architectures and different downstream datasets and tasks to achieve performance gain. Currently the size of the real cooperation dataset is relatively small and it will be interesting if larger cooperative datasets can be collected for pre-training in the future.

ACKNOWLEDGMENTS

Ping Luo is supported by the General Research Fund of HK No.27208720, No.17212120, and No.17200622.

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

## A    BACKGROUND ABOUT VIEW BUILDING IN CONTRASTIVE LEARNING

In this section, we discuss how to build proper views in contrastive learning. Firstly, we introduce the formulation of contrastive learning and some important properties of good views in contrastive learning as proposed in (Tian et al., 2020). Then we discuss view building for LiDAR point clouds.

**View Building Contrastive Learning.** Unsupervised representation learning aims to pre-train 2D/3D backbones on a dataset without labels, which can be transferred to downstream datasets and tasks to achieve performance improvement over training from scratch (random initialization). Recently, unsupervised contrastive learning achieves great success in image domain (He et al., 2020; Tian et al., 2019; Caron et al., 2020; Grill et al., 2020; Wang et al., 2021). Given a batch of images $\mathcal{X}$ as inputs, these works first apply two kinds of random augmentations for each image $x^n \in \mathcal{X}$ ($n = 1, 2, ..., N$ where $N$ is the number of images in the batch) to get augmented images $x_1^n$ and $x_2^n$, which are called different views of $x^n$. The main objective of contrastive learning is to pull together the representations of views of the same image in the feature space while pushing away representations of different images, as indicated in the equation below:

$$L_{\text{con}} = \frac{1}{N} \sum_{n=1}^{N} -\log(\frac{\exp(z_1^n \cdot z_2^n / \tau)}{\sum_{i=1}^{N} \exp(z_1^n \cdot z_2^i / \tau)}) \quad \text{with}$$
$$z_{1,2}^n = f_I^{\text{enc}}(x_{1,2}^n) \quad n = 1, 2, ..., N \tag{8}$$

where $f_I^{\text{enc}}$ is the 2D backbone used to extract representations. $z_{1,2}^n$ is the encoded representations for views $x_{1,2}^n$. To apply contrastive loss in the first line in Eqn. (8), views of the same image are considered as positive pairs and other pairs are negative ones. The numerator indicates the similarity of positive pairs while the denominator sums up the positive similarity and the sum of similarity of negative pairs. $\tau$ is the temperature parameter. Minimizing this loss equals to maximize the similarity of positive pairs and minimize the similarity of negative pairs.

Authors in (Tian et al., 2020) discuss what property views should have to benefit contrastive learning via Information Theory and propose that mutual information (Shannon, 2001; Kreer, 1957; Wikipedia contributors, 2022) of views ,$I(x_1^n; x_2^n)$, can indicate the quality of the learned representations. Mutual information formally quantifies "how much information about one random variable we can obtain when observing the other one". Experiments on images suggest that there exists a "sweet spot" for $I(x_1^n; x_2^n)$ where the pre-trained backbone can achieve the most significant performance improvement in downstream tasks. This means the mutual information between views can neither be too low (sharing little semantics) nor too high (differing little). Further experiments in (Tian et al., 2020) indicate that the mutual information of different augmented images is high and reducing $I(x_1^n; x_2^n)$ by applying stronger augmentations is effective. The performance on downstream tasks increases at the beginning and then decrease when the augmentations are too strong.

**Views Building for LiDAR Point Clouds.** As discussed in the main paper, it is impossible for us to reconstruct the whole outdoor-scene for constrative learning, which is demonstrated useful in indoor-scene (Xie et al., 2020; Hou et al., 2021; Liu et al., 2020). And there exists two alternatives to build views for outdoor-scene LiDAR point clouds. The first one (Liang et al., 2021; Yin et al., 2022) is to apply data augmentation to single frame of point cloud and treat the original and augmented versions as different views, which is similar to what previous works do in image domain. However, the augmentations in image domain are highly non-linear while all the augmentation of point clouds, including random drop, rotation and scaling, can be implemented in a linear transformation. As claimed in (Tian et al., 2020), the highly non-linear augmentations on images already bring high mutual information between views. Thus views of LiDAR point clouds built in this way would have higher mutual information, which is not adequate for learning representations. The second intuitive idea to build views is to utilize point clouds at different timestamps, embraced by (Huang et al., 2021). However, outdoor-scenes are dynamic and the autonomous driving vehicle has no idea about how other objects (cars, pedestrians, etc.) move. Thus observing one view (timestamp t) bring in little information about the other one (timestamp t+10 for example), meaning that $I(x_1^n; x_2^n)$ can be extremely low and this can be harmful to the learned representations. Due to these limitations, pre-trained 3D encoders in (Liang et al., 2021; Huang et al., 2021) cannot achieve noticeable improvement when transferring to datasets collected by different LiDAR sensors. Thus, in this paper, we propose to utilize the vehicle-infrastructure-cooperation dataset (Yu et al., 2022), which capture the same scene from different view-point at the same timestamp, for contrastive representation learning. Views built

with this dataset neither share too much information (captured from different view-points) nor share too little information (captured at the same time, easy to find correspondence), which is adequate for contrastive learning.

# B    RECONSTRUCTION OBJECTIVE FOR TASK-RELEVANT INFORMATION

In this section, we borrow the ideas from (Wang et al., 2022) to explain why pure contrastive learning bring less improvements as shown in Table 3 in the main paper. Firstly, we give the definition of sufficient representation and minimal sufficient representation in contrastive learning. Then, we present analysis on image classification problem as downstream task and we refer readers to (Wang et al., 2022) for other downstream tasks. Finally, we propose our pre-training objective for LiDAR point clouds.

**Sufficient Representation and Minimal Sufficient Representation.** Sufficient Representation $z_{1,\text{suf}}^n$ of $x_1^n$ contains all the information that is shared by $x_1^n$ and $x_2^n$, which means $z_{1,\text{suf}}^n$ can be used to express common semantics shared by these two views. Among all the sufficient representations for $x_2^n$, minimal sufficient representation $z_{1,\text{min}}^n$ contains the least information about $x_1^n$. The learned representation in contrastive learning is sufficient and almost minimal. Assuming that $\mathcal{Z}_{1,\text{suf}}$ is the set of all possible sufficient representations of view $x_1^n$, we can define these two concepts as belows,

**Definition 1.** $z_{1,suf}^n$ of view $x_1^n$ is sufficient for $x_2^n$ **if and only if** $I(z_{1,suf}^n, x_2^n) = I(x_1^n, x_2^n)$.

**Definition 2.** $z_{1,min}^n \in \mathcal{Z}_{1,suf}$ of view $x_1^n$ is minimal sufficient **if and only if** $I(z_{1,min}^n, x_1^n) \leq I(z_{1,suf}^n, x_1^n), \forall z_{1,suf}^n \in \mathcal{Z}_{1,suf}$.

**Theorem.** (1) $z_{1,suf}$ provides more information about the downstream task $T$ than $z_{1,\text{min}}$. (2) The upper bound of error rates in downstream tasks using minimal sufficient representations are higher than that of sufficient representations. That is,

$$I(z_{1,\text{suf}}, T) \geq I(z_{1,\text{min}}, T)$$
$$\sup\{P_{\text{suf}}^e\} \leq \sup\{P_{\text{min}}^e\} \tag{9}$$

This gap stems from the missing task-relevant information in $z_{1,\text{min}}$. To prevent this problem, authors in (Wang et al., 2022) propose to add a reconstruction objective (reconstruct $x_1$ using $z_1$) alongside the contrastive loss to increase $I(z_1, x_1)$, which indirectly increases $I(z_1, T|x_2)$ and brings improvement in downstream classification problem over pure contrastive learning.

**Proof for Image Classification Problem.** We denote the downstream classification task as $T$ and the task-relevant information in minimal sufficient representation $z_{1,\text{min}}$ can be described as below:

$$I(z_{1,\text{suf}}, T) = I(z_{1,\text{min}}, T) + [I(x_1, T|z_{1,\text{min}}) - I(x_1, T|z_{1,\text{suf}})]$$
$$\geq I(z_{1,\text{min}}, T) \tag{10}$$

To begin with, $z_{1,\text{suf}}$ and $z_{1,\text{min}}$ are sufficient representations and they contain two parts of information: shared information between $x_1$ and $x_2$, and extra information about $x_1$. Thus the mutual information $I(z_{1,\text{suf}}, T)$ can be decomposed into $I(z_{1,\text{min}}, T)$ and $[I(x_1, T|z_{1,\text{min}}) - I(x_1, T|z_{1,\text{suf}})]$, where $I(x_1, T|z_{1,\text{min}})$ indicates the information about $T$ we can obtain by observing $x_1$ when $z_{1,\text{min}}$ is known. As $z_{1,\text{suf}}$ contains more information about $x_1$ than $z_{1,\text{min}}$, the second term is larger than zero and the right-hand-side of the first line in Eqn. (10) is larger than $I(z_{1,\text{min}}, T)$. This indicates that $z_{1,\text{suf}}$ contains more task-relevant information than $z_{1,\text{min}}$ and thus would have better performance in $T$. Then we consider using Bayes error rate $P^e$ (Fukunaga, 2013), which is the lower-bound of achievable error for the classifier, to analyze performance of $z_{1,\text{suf}}$ and $z_{1,\text{min}}$ on downstream classification problem. We have

$$P_{\text{suf}}^e \leq 1 - \exp[-H(T) + I(x_1, x_2, T) + I(z_{1,\text{suf}}, T|x_2)]$$
$$P_{\text{min}}^e \leq 1 - \exp[-H(T) + I(x_1, x_2, T)] \tag{11}$$

where $H(T)$ is the entropy of the task. Since $1 - \exp[-(H(T) + I(x_1, x_2, T) + I(z_{1,\text{suf}}, T|x_2)] \leq 1 - \exp[-H(T) + I(x_1, x_2, T)]$, the upper bound of Bayes error rate of minimal sufficient representation is larger than that of sufficient representations. This indicates that ideally $z_{1,\text{suf}}$ can achieve better performance than $z_{1,\text{min}}$ in classification problem.

**Contextual Shape Prediction Objective.** As it is impossible to reconstruct the whole scene point cloud with point-level or voxel-level representations. We propose an additional pre-training objective to predict distribution of local neighborhood using point/voxel-level representation. We use shape context to describe the local neighborhood distribution of a point/voxel, which has been demonstrated as a useful local distribution descriptor in previous works (Hou et al., 2021; Belongie et al., 2002; Körtgen et al., 2003; Xie et al., 2018). As demonstrated in our ablation study (Table 3 in main paper), this additional pre-training objective bring more significant performance improvement over pre-trained by pure contrastive loss. We also provide python-style code for computing shape context as followings

---

**Algorithm 1** Implementation of Contextual Shape Computation in Python Style.

```python
class Contextual_Shape(object):
    def __init__(self, r1=0.125, r2=2, nbins_xy=2, nbins_zy=2):
        self.r1 = r1
        self.r2 = r2
        self.nbins_xy = nbins_xy
        self.nbins_zy = nbins_zy
        self.partitions = nbins_xy * nbins_zy * 2

    def pdist_batch(rel_trans):
        D2 = torch.sum(rel_trans.pow(2), 3)
        return torch.sqrt(D2 + 1e-7)

    def compute_rel_trans_batch(A, B):
        return A.unsqueeze(1) - B.unsqueeze(2)

    def hash_batch(A, B, seed):
        mask = (A >= 0) & (B >= 0)
        C = torch.zeros_like(A) - 1
        C[mask] = A[mask] * seed + B[mask]
        return C

    def compute_angles_batch(rel_trans):
        angles_xy = torch.atan2(rel_trans[:, :, :, 1], rel_trans[:, :, :, 0])
        angles_xy = torch.fmod(angles_xy + 2 * math.pi, 2 * math.pi)
        angles_zy = torch.atan2(rel_trans[:, :, :, 1], rel_trans[:, :, :, 2])
        angles_zy = torch.fmod(angles_zy + 2 * math.pi, math.pi)
        return angles_xy, angles_zy

    def compute_partitions_batch(self, xyz_batch):
        rel_trans_batch = ShapeContext.compute_rel_trans_batch(xyz_batch,
            xyz_batch)
        # compute angles from different points to the query one
        angles_xy_batch, angles_zy_batch = ShapeContext.compute_angles_batch(
            rel_trans_batch)
        angles_xy_bins_batch = torch.floor(angles_xy_batch / (2 * math.pi /
            self.nbins_xy))
        angles_zy_bins_batch = torch.floor(angles_zy_batch / (math.pi / self.
            nbins_zy))
        angles_bins_batch = ShapeContext.hash_batch(angles_xy_bins_batch,
            angles_zy_bins_batch, self.nbins_zy)
        # compute distances between different points and the query one
        distance_matrix_batch = ShapeContext.pdist_batch(rel_trans_batch)
        dist_bins_batch = torch.zeros_like(angles_bins_batch) - 1
        # generate partitions for each points
        mask_batch = (distance_matrix_batch >= self.r1) & (
            distance_matrix_batch < self.r2)
        dist_bins_batch[mask_batch] = 0
        mask_batch = distance_matrix_batch >= self.r2
        dist_bins_batch[mask_batch] = 1
        bins_batch = ShapeContext.hash_batch(dist_bins_batch,
            angles_bins_batch, self.nbins_xy * self.nbins_zy)
        return bins_batch
```

---

## C  DATASETS DETAILS

In this section, we introduce details about different datasets used in the main paper for evaluation and also two more datasets in additional experiments.

**DAIR-V2X.** *DAIR-V2X* (Yu et al., 2022) is the first real-world autonomous dataset for vehicle-infrastructure-cooperative detection task. It covers various scenes, including cities and highways, and different whether condition including sunny, rainy and foggy days. A virtual world coordinate is used to align the vehicle LiDAR coordinate and infrastructure LiDAR coordinate. There are 38845 LiDAR frames (10084 in vehicle-side and 22325 in infrastructure-side) for cooperative-detection task. The dataset contains around 7000 synchronized cooperative samples in total and we utilize them to pre-train 3D encoder in an unsupervised manner via the proposed CO3. The LiDAR sensor at vehicle-side is 40-beam while a 120-beam LiDAR is utilized at infrastructure-side.

**Once.** *Once* (Mao et al., 2021) is a large-scale autonomous dataset for evaluating self-supervised methods with 1 Million LiDAR frames and only 15k fully annotated frames with 3 classes (Vehicle, Pedestrian, Cyclist). A 40-beam LiDAR is used in (Mao et al., 2021) to collect the point cloud data. We adopt common practice, including point cloud range and voxel size, in their public code repository[3]. As for the evaluation metrics, IoU thresholds 0.7, 0.3, 0.5 are respectively adopted for vehicle, pedestrian, cyclist. Then 50 score thresholds with the recall rates ranging from 0.02 to 1.00 (step size if 0.02) are computed and the 50 corresponding values are used to draw a PR curve, resulting in the final mAPs (mean accurate precisions) for each category. We also further overage over the three categories and compute an 'Overall' mAP for evaluations.

**KITTI.** *KITTI* (Geiger et al., 2012) is a widely used self-driving dataset, where point clouds are collected by LiDAR with 64 beams. It contains around 7k samples for training and another 7k for evaluation. For point cloud range and voxel size, we adopt common practice in current popular codebase like MMDetection3D[4] and OpenPCDet[5]. All the results are evaluated by mAPs with three difficulty levels: Easy, Moderate and Hard. These three results are further average and an 'Overall' mAP is generated for comparisons.

## D  ADDITIONAL EXPERIMENT RESULTS

### D.1  INFRASTRUCTURE AS VIEW IN CONTRASTIVE LEARNING

We also conduct ablation study on the fusion view. Instead of using fusion point clouds as view in contrastive learning, we directly use infrastructure side point cloud as another view. Results are shown in Table 7. It can be found that if we directly use infrastructure-side point cloud for contrastive learning, the performance improvement is very marginal. This stems from the sparsity of LiDAR point clouds, which sometimes make it difficult to find good positive pairs to perform contrastive learning.

| Init. | Overall | Vehicle | Pedestrian | Cyclist |
|---|---|---|---|---|
| Random | 55.92 | 62.85 | 45.52 | 59.39 |
| Inf-view | 56.45 | 62.71 | 46.63 | 60.02 |
| CO3 | 58.50 | 64.60 | 48.83 | 62.17 |

Table 7: Resuls of directly using infrastructure-side point cloud as view in contrastive learning.

## E  IMPLEMENTATION DETAILS

In this section, we introduce some details about implementation in both pre-training stage and fine-tuning stage. Common settings of pre-training and fine-tuning are listed in Table 8 and we discuss other settings that vary in different detectors later.

---

[3]https://github.com/PointsCoder/Once_Benchmark
[4]https://github.com/open-mmlab/mmdetection3d
[5]https://github.com/open-mmlab/OpenPCDet

| Configuration | Pre-training | KITTI | Once |
|---|---|---|---|
| optimizer | AdamW | Adam | Adam |
| base learning rate | 0.0001 | 0.003 | 0.003 |
| weight decay | 0.01 | 0.01 | 0.01 |
| batch size | 16 | - | - |
| learning rate schedule | cyclic | cyclic | cyclic |
| GPU numbers | 8 | 4 | 4 |
| training epochs | ***20*** | 80 | 80 |

Table 8: Details about implementations. "Pre-training" means settings in *DAIR-V2X* (Yu et al., 2022). "KITTI" and "Once" respectively indicate settings for detection tasks in *KITTI* (Geiger et al., 2012) and *Once* (Mao et al., 2021). We list all common settings and discuss those vary in different detectors below, which are marked as "-" in this table.

**Other Pre-training Settings.** To accelerate the pre-training process, we utilize the "repeated dataset" in MMDetection3D and the schedule is set to 10 epochs, which equals to 20 epochs without "repeated dataset". Thus the number 20 for training epochs in Table 8 is tilt.

**Other Fine-tuning Settings.** Batch size settings for different detectors on different datasets are shown in Table 9. We use different types of GPUs, different number of GPUs and different version of PyTorch (Paszke et al., 2019) as compared to those used in the codebases, which may lead to degrading when training from scratch. Thus these parameters are tuned based on the original settings from the codebases to make the performance of training from scratch match or even surpass the results they published.

| Detectors | KITTI | Once |
|---|---|---|
| Second | 48 | 48 |
| PV-RCNN | 16 | 48 |
| CenterPoint | - | 32 |

Table 9: Details about batchsize settings for different detectors on different datasets. "-" means there is no configuration for the detector on the exact dataset in the codebase or we do not conduct the downstream experiments.

## F   TABLE OF NOTATION

| Variable | Description | Note |
|---|---|---|
| $\mathbf{P}_{v/f}$ | Raw vehicle/fusion LiDAR points. | |
| $\hat{\mathbf{P}}_{v/f}$ | Point/Voxel-level features after embedded by the 3D backbone. | Indicating same process applied to vehicle/fusion point cloud. |
| $\hat{\mathbf{Z}}_{v/f}$ | Projected point/voxel-level features in the common feature space. | |
| $\mathbf{Z}_{v/f}$ | Normalized point/voxel-level features in the common feature space. | An L-2 normalization is applied on the feature dimension of $\hat{\mathbf{Z}}_{v/f}$. $z_{v/f}^n$ is sampled point/voxel feature for contrastive learning. |
| $N_1$ | The number of samples for contrastive learning. | |
| $\tau$ | Temperature parameter in contrastive learning. | |
| $Q'$ | A matrix in dimension $\mathbb{R}^{N_f \times N_{bin}}$ indicating the number of points in each bin in the neighborhood of each fusion point. | |
| $Q$ | Apply $\ell_2$-normalization and softmax to $Q'$ then we get $Q \in \mathbb{R}^{N_f \times N_{bin}}$, which describes a local geometry distribution in the neighborhood of each fusion point. | $q_{v/f}^n$ is sampled "ground truth" distribution from Q |
| $P$ | Predicted local geometry distribution from $\mathbf{P}_{v/f}^{enc}$. | $p_{v/f}^n$ is sampled distribution prediction from P. |
| $N_2$ | The number of samples for contextual shape prediction. | |

Table 10: Detailed description of variables

## G   DISCUSSION ABOUT THE ABLATION STUDY

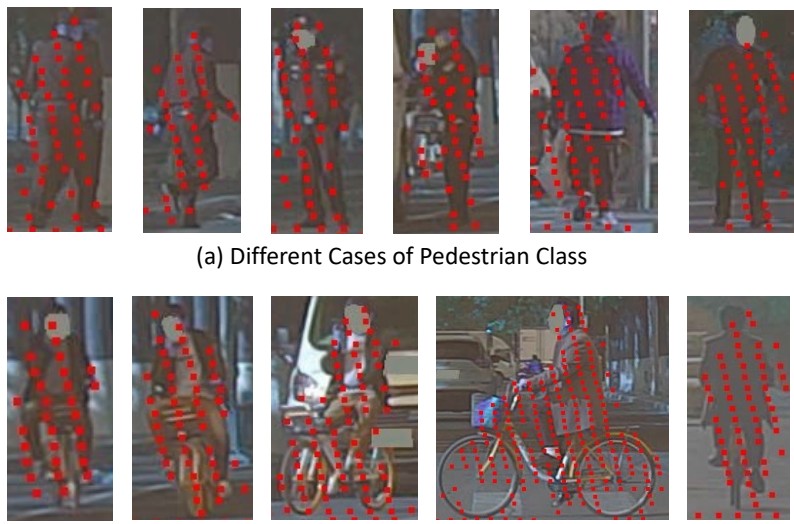

(a) Different Cases of Pedestrian Class

(b) Different Cases of Cyclist Class

Figure 6: Example samples of pedestrian and cyclist. LiDAR points are shown in red. It can be found that pedestrian category has consistent shape while the shape of cyclist varies across time and identities.

It can be found in Table 6 that using contextual shape prediction loss only brings more improvement on Pedestrian class while cooperative contrastive loss introduce more gains on the Cyclist class. In this section, we provide a discussion on this phenomenon.

First of all, the "ground truth" shape context is computed with point clouds and the contextual shape prediction goal is to predict the local point distribution with voxel-level representations. This

enables the voxel-level representation to predict the structure inside a voxel. Meanwhile, cooperative contrastive only focus on contrast different voxel representation. For pedestrian class, there are usually only one or two voxels for one pedestrian. Thus, with cooperative contrastive loss only, the representation fail to recognize the inner structure of the voxel and thus cannot improve upon the pedestrian category. On the contrary, when we pre-train the encoder with contextual shape prediction loss only, the learned representation is able to express the structure inside the voxel and this helps with the downstream detection on the pedestrian category.

Second, as the contextual shape prediction goal aims to capture local shape distribution, it fails to learn good representation with varying shape. Compared to pedestrians whose shape is always cylinder-like, cyclists with their bicycles are usually captured with different poses, leading to different shapes. A visualization of different cases of the two categories are provided in Fig. 6. Thus the representation fails to learn knowledge when predicting varying shape of the same semantic. When we look at the cyclist category, it can be found that pre-training with contextual shape loss only brings little improvement.

Besides, as discussed in Appendix B, the two pre-training objectives are complementary. Pure cooperative contrastive learning makes the representation minimal sufficient, which lacks of task-relevant information. The contextual shape prediction loss brings more task-relevant information by increasing the mutual information between the representations and the inputs. Thus combining them leads to better performance.

## H  MORE EXPERIMENTS.

### H.1  ABLATION EXPERIMENTS ON SHAPE CONTEXT.

We conduct a ablation experiment on the contextual shape prediction loss. Previous works (Belongie et al., 2002; Körtgen et al., 2003) use the number of points in each bin as the feature of the query point. However, we think it is difficult for the network to directly regress the exact number of points in the bins. Thus we propose to construction a local distribution and use the representation to predict it. In this part, we conduct ablation study on this design. We use the exact number of points as prediction goal and keep other parts of our method the same. The experiment results are shown in Table 11 and it can be found that directly transfer the idea of shape context only brings little improvement (0.94mAP) as compared to the proposed CO3 (2.58 mAP). Thus our insight can also be transferred to indoor scene point clouds.

| Initialization | Overall mAP | Vehicle | Pedestrian | Cyclist |
|---|---|---|---|---|
| Random | 55.92 | 62.85 | 45.52 | 59.39 |
| CO3 with exact number of points prediction | 56.86 (+0.94) | 62.79 | 47.63 | 60.15 |
| CO3 | 58.50 (+2.58) | 64.60 | 48.83 | 62.17 |

Table 11: Ablation experiments on shape context.

### H.2  EXPERIMENTS COMPARED TO SAFETY SPACE FORECASTING.

We also conduct experiments where freespace forecasting (Hu et al., 2021) is used as pre-training goal. The pre-training setting is kept the same as all the other initialization methods. The results are shown in Table 12. It can be found that pre-training with safety space forecasting (Hu et al., 2021) brings minor performance gain. We think this might stems from the objective of the pre-training goal. The pre-training goal of (Hu et al., 2021) is to predict freespace for safe driving and their downstream task is motion planning. However, 3D perception task requires semantic information and simply predicting freespace does not help to distinguish the representation of different objects. This is why pre-training with freespace forecasting bring lower performance improvement on 3D perception task.

### H.3  PARAMETER SENSITIVITY EXPERIMENTS

In this part, we conduct parameter sensitivity experiments on temperature parameter and radius in contextual shape prediction loss.

| Initialization | Overall mAP | Vehicle | Pedestrian | Cyclist |
|---|---|---|---|---|
| From Scratch | 55.92 | 62.85 | 45.52 | 59.39 |
| Freespace forecasting | 56.18 (+0.26) | 63.33 | 46.19 | 59.01 |
| CO3 | 58.50 (+2.58) | 64.60 | 48.83 | 62.17 |

Table 12: Experiment results on freespace forecasting pre-training.

We first fix all the other parameters in the main paper and change $R1$ and $R2$ in the contextual shape prediction loss. We pre-train the 3D encoder on the DAIR-V2X dataset and downstream it to Once dataset with CenterPoint as detector. Results are shown in Table 13. It can be found that for the same R2, an increasing R1 brings more improvement on the downstream detection tasks. This might stem from that a larger inner radius can help capture more neighborhood information, making the contextual shape prediction goal more meaningful. It can also be found that when $R2$ is relatively small, for example R2=3.5m and R1=1m, the performance drops. This might stem from the relatively small R2 which might fail to capture the local shape distribution of a larger object like cars or trucks. Also, we did not search these parameters before and we surprisingly find that using R1=1.5m and R2=4m brings the best performance (58.86 mAP).

| R2/m \ R1/m | 0.5 | 1.0 | 1.5 | 2.0 |
|---|---|---|---|---|
| 3.5 | 57.55 | 57.31 | 58.00 | 58.42 |
| 4.0 | 57.53 | 58.50 | 58.86 | 58.48 |

Table 13: Results on parameter sensitivity experiments on temperature.

In the experiment on temperature, all other parameters are kept the same as those in the main experiment and we only change the temperature in the cooperative contrastive loss. We pre-train the 3D backbone on DAIR-V2X dataset with different parameter settings and fine-tune it on Once dataset with CenterPoint. The results are shown in Table 14. It can be found that as the temperature increases to 0.15, the downstream performance drop to 57.34 mAP. This is because a higher temperature brings smaller gap between negative pairs and this makes the representations hard to separate different objects. Also, as we keep decreasing the temperature, the overall downstream performance drop to 57.74 mAP. This is because too small temperature push representation of different objects too far away but ignore the similar semantic meaning for the same category. For example, it will make representation of two different cars far away and this will harm the performance of downstream detection task. It can also be found that using adequate temperature brings comparable performance (58.50 mAP vs 58.10 mAP).

| Temperature | 0.02 | 0.07 | 0.1 | 0.15 |
|---|---|---|---|---|
| Overall mAP | 57.74 | 58.50 | 58.10 | 57.34 |

Table 14: Results on parameter sensitivity experiments on radius in contextual shape prediction loss.

# I  DISCUSSION ON THE INFLUENCE TO V2X COMMUNITY.

The V2X community is developing rapidly and the original motivation of V2X setting is to alleviate occlusion and long-range sensing problems (Yu et al., 2022). There are several V2X settings including vehicle-to-infrastructure and vehicle-to-vehicle.

In this paper, we propose to use cooperation dataset for 3D unsupervised representation learning and achieve performance improvement on perception task using vehicle-side point clouds only. Although our experiments are conducted on vehicle-infrastructure dataset, our method can also be applied on other V2X settings without any label. As labeling is the most intensive and time-consuming part in the collection of cooperation dataset, we believe larger scale of unlabeled cooperation dataset will be collected in the future for unsupervised 3D representation learning to introduce more performance gain. Also, it is expensive and difficult to deploy V2X settings everywhere. In our work, we pre-train on such cooperation dataset and achieve improvements on downstream tasks with vehicle-side

point clouds as inputs, which is a new exciting finding about the V2X research. We believe the promising results will encourage more attempts in cooperative unsupervised representation learning and accelerate the development of the V2X community.

