# OpenReview forum: "CO3: Cooperative Unsupervised 3D Representation Learning for Autonomous Driving"
_ICLR.cc/2023/Conference — ICLR 2023 poster_

### Official Review · Reviewer_bVKL · 2022-10-23

**Confidence:** 5
**Correctness:** 3
**Technical Novelty And Significance:** 3
**Empirical Novelty And Significance:** 2
**Recommendation:** 8

**Clarity, Quality, Novelty And Reproducibility:**

Clarity and Quality: As mentioned above, the clarity and the quality of the paper are high

Novelty: I believe the amount of novelty is sufficient for publication, if we consider the combination of using a new source of data for unsupervised representation learning for LIDAR point cloud and the use of a task-relevant loss based on shape

Reproducibility: Sufficient implementation details are provided. However, the paper uses quite a number of hyperparameters (e.g. temperature, shape descriptor radii, etc.), and it is unclear how sensitive the method is to the values of these hyperparameters

**Strength And Weaknesses:**

Strengths:

The core ideas in the paper are interesting. I liked the concept of introducing a more task-relevant loss based on shape in the self-supervised stage
The evaluation procedure is mostly thorough and sound, using several public benchmarks, comparing against relevant related work, and with appropriate ablative analysis. Results are consistent across datasets, which is convincing (see however my concern under weaknesses on the magnitude of the deltas)
The paper is written with a level of clarity that is significantly above average (I commend the authors for this, it makes reviewing so much easier!)
Weaknesses:

In my opinion, the main weakness is the magnitude of the improvements. I am looking for example at Table 2, where the difference between the various methods in <0.5 mAP. It would be valuable to have some sort of estimate of the variance, for example, how much does performance change upon different runs of training (e.g. with different random initialisation) and (less important) across the values of hyperparameters (e.g. the temperature, the shape descriptor radii) . However, it has to be said that, while small, the improvement can be observed consistently across various datasets, tasks, and networks used for supervised learning
The method to generate data for self-supervised training seems hard to scale, as it seems quite expensive and operationally hard to get to work (collaborative infra and vehicles), which might be the reason why the cooperation dataset is fairly small. Related to this, could the authors comment on the sensitivity of the method to the accuracy of synchronisation and relative point estimation between the vehicle and the infra point cloud? How is the relative pose computed?
The paper needs a more in depth overview of related work on two fronts: use of task-relevant losses as part of the self-supervised learning, and use of shape descriptors like the one in Sec. 3.3 in the context of learning from point clouds.
Minor details for the authors for improvements in the next version of the paper:

I would recommend to bring Table 7 from the Appendix into the main paper
It would be useful to add some rationale or ablative analysis on the choice of some parameters (e.g. the radii of the shape descriptors (Sec. 3.3)
It is hard to understand/see the negative and positive pairs in Fig. 3
The math notation is a bit cumbersome. For example, it took me a while to understand the v/f notation
I have some questions on invariance to rotations: is the shape descriptor rotationally invariant? If yes, how does it work? Or it does not matter because the infra and vehicle point clouds are aligned first? If the point clouds are aligned, how do methods learn about rotational invariance: are there such kind of augmentations in the supervised learning stage?
Some statements are fairly strong, such as: "This stems from the sparsity of LiDAR point clouds, which sometimes make it difficult to find good positive pairs to perform contrastive learning". Unless there is some strong evidence backing this claim, I would recommend it softening, e.g. "We believe this stems from the sparsity..."

**Summary Of The Paper:**

This paper proposes a method for self-supervised representation learning of LIDAR point clouds, to initialise training of standard supervised methods for LIDAR 3D object detection and point cloud segmentation. The paper has two main contributions:

The self-supervised representation is learnt via contrastive learning on point clouds from LIDAR mounted on vehicles and LIDAR mounted on infrastructure, synchronised with respect to each other. This provides more diversity than state-of-the-art methods, that use ad-hoc augmentations or LIDAR point clouds collected by the same vehicle but at different timestamps (which is problematic due to moving objects)
A shape-based loss as part of the unsupervised learning step that is meant to bring in task-specific information into this step
The paper is evaluated on standard 3D object detection and LIDAR semantic segmentation benchmarks, showing marginal (but consistent) improvements over alternative self-supervised initialisation methods.

**Summary Of The Review:**

Overall, I recommend accepting the paper, and I'd like to invite the authors to address the points I listed in the weaknesses section.

There is enough novelty in the contributions (using a new source of point cloud data for self-supervised representation learning for point clouds and use of task-relevant losses based on shape during self-supervised learning). There is also enough signal showing the usefulness of the method. While the gains compared to previous methods are overall small, they are very consistent across datasets, supervised tasks, and supervised methods. Some additional analysis on the sensitivity to multiple runs (to get an estimate of the variance) would make the case stronger. The paper is clearly explained, with enough details for reproducibiltiy.

---

> ### Author Response · Authors · 2022-11-09
> **Response to Reviewer bVKL.**
>
> Dear Reviewer bVKL,
>
> Many thanks for your time on the review and the constructive suggestions! We provide discussion about the magnitude of performance improvement and conduct repeated experiment results on KITTI in the [General Response Part 3](https://openreview.net/forum?id=QUaDoIdgo0&noteId=Ew-KG93lBk). We also discuss how the cooperation datset is collected and how relative poses are computed. Besides we provide overview of shape context and reconstruction goal in the related work part. We also revise our presentation according to your constructive suggestions.
>
> **Q1:** "In my opinion, the main weakness is the magnitude of the improvements. I am looking for example at Table 2 ... However, it has to be said that, while small, the improvement can be observed consistently across various datasets, tasks, and networks used for supervised learning "
>
> **A1:** We provide an discussion on the performance improvement, especially on PV-RCNN and KITTI dataset. We also conduct repeated experiments on KITTI dataset in  [General Response Part 3](https://openreview.net/forum?id=QUaDoIdgo0&noteId=Ew-KG93lBk). We provide a brief summary here.
>
> The relatively small improvement on PV-RCNN stems from that it uses two 3D encoder while only the voxel-based encoder is pre-trained. Similar lower improvement can be observed in other initialization methods. When we look at PV-RCNN on Once dataset, it can be found that CO3 achieves the best performance among various initialization methods and 100% more improvement than the second place. Some other initialization methods even harm the performance of PV-RCNN.
>
> As for the KITTI dataset, the performance is already high and this makes the improvement seemingly marginal. In order to alleviate the concern, we conduct repeated experiments on KITTI with Voxel-RCNN (enabling faster training). We train it for 5 times on KITTI and compute the mean and variance of the results. Results are shown in Table 2 in the  [General Response Part 3](https://openreview.net/forum?id=QUaDoIdgo0&noteId=Ew-KG93lBk). It can be found that the improvement is not about randomness and CO3 achieves up to 300% improvement than other methods.
>
> **Q2:** "The method to generate data for self-supervised training seems hard to scale, as it seems quite expensive ... How is the relative pose computed? "
>
> **A2:** Point clouds in both sides are collected with respective timestamps and relative poses in the world coordinate. Timestamps are from the collecting system itself and poses are computed by an IMU&GPU SLAM system with some post-processing. Then the synchronization is conducted based on the timestamps. Pointclouds from both sides with timestamp difference less than 0.03s are considered as one pair in the cooperaitve dataset. Once we have a pair of pointclouds together with their poses in the world coordinate, we can compute the relative poses. The process of synchronization and pose computation can be done easily with the collected data.
>
> The scale of the DAIR-V2X dataset is relatively small because it aims for detection task and requires labels. Labeling on such dataset is much more intensive than labeling on normal autonomous driving dataset, which limit its scale. However, our proposed method does not require labels and as described above, the synchronization process and computation of relative pose can be done with similar cost of normal autonomous driving dataset (they also collect timestamps and poses in the world coordinate). Thus we think in the future, a larger scale unlabelled cooperation dataset can be collected and used for pre-training to bring more performance gain.
>
>
> **Q3:** "The paper needs a more in depth overview of related work on two fronts: use of task-relevant losses as part of the self-supervised learning, and use of shape descriptors like the one in Sec. 3.3 in the context of learning from point clouds. "
>
> **A3:** Thank you for your suggestion on improving our presentation! We add the discussion of task-relevant losses and contextual shape descriptor in the related works part in the revision (highlighted in blue).
>
> **Task-relevant information.** As discussed in [1] and Appendix B in our paper, pure contrastive learning forces the learned represetation to be minimal sufficient, which lacks of task-relevant information. To inject more task-relevant information, a reconstruction goal can be used alongside the contrastive learning goal. Analysis and proof in a simpler scene where the task is image classification is also provided in Appendix B.
>
> **Shape context.** Previous works [2,3,4,5] divides the local neighborhood of a point into several bins and compute the number of points in each bin, which is used as the shape context descriptor for the point and is demonstrated useful in other applications such as point cloud registration. Instead of predicting exact number of points, we propose to construct a local distribution upon shape context and achieve remarkable performance gain.

---

> > ### Author Response · Authors · 2022-11-09
> > **Response to "Other details for improving the paper". Thank you!**
> >
> > **Other details for improving the paper.**
> >
> > * **About the location of Table 7.** We believe that the expression can be clearer if we move Table 7 to the main paper. However, the space is limited and the table occupies some space. Instead of directly moving it to the main paper, we add the performance gain comparison in the text in Section 3.3 in the revision (highlighted in blue) to make it clearer. We will try our best to rearrange the paper in the final version to bring it to the main paper.
> > * **About the parameter sensitivity experiments.** We are running sensitivity experiments on parameters like temperature and radius in the shape context. We will update the results as soon as the experiments finish. Thank you for the suggestion.
> > * **About the neg/pos pairs visualization in Figure 3.** We revise Figure 3 and add circles to indicate the positive and negative pairs in the scene. Thank you for the suggestion.
> > * **The math notation about v/f** We are sorry to confuse you. We use subscript 'v/f' to indicate that the same operation is applied respectively on both pointclouds. We have added description about the subscript 'v/f' before and after the Equation (1), highlighted in blue in the revision. We hope that can make our presentation clearer. Thank you.
> > * **About the rotation-invariant property** In the pre-training stage, we first align point clouds from both sides and apply the same point cloud augmentation including random rotation to them before passing them into the 3D encoder. Also in the finetuning stage, random rotation is one of the augmentation methods. These are how we ensure rotation-invariance.
> > * **About the strong claims in the paper.** Thank you for pointing this out! It should be better to use "we think that ...". We have revised some claims in the revision and highlight them in blue.
> >
> >
> > We hope our reply can help you better understand our paper and alleviate your concern. Look forward to further discussion!
> >
> > Best regards,
> > Authors of Paper 1639
> >
> >
> > [1] Haoqing Wang, Xun Guo, Zhi-Hong Deng, and Yan Lu. Rethinking minimal sufficient representation in contrastive learning. arXiv preprint arXiv:2203.07004, 2022.
> >
> > [2] Ji Hou, Benjamin Graham, Matthias Nie.ner, and Saining Xie. Exploring data-efficient 3d scene understanding with contrastive scene contexts. In Proceedings of the IEEE/CVF Conference on Computer Vision and Pattern Recognition, pp. 15587–15597, 2021.
> >
> > [3] Serge Belongie, Jitendra Malik, and Jan Puzicha. Shape matching and object recognition using shape contexts. IEEE transactions on pattern analysis and machine intelligence, 24(4):509–522, 2002.
> >
> > [4] Marcel K.rtgen, Gil-Joo Park, Marcin Novotni, and Reinhard Klein. 3d shape matching with 3d shape contexts. In The 7th central European seminar on computer graphics, volume 3, pp. 5–17. Citeseer, 2003.
> >
> > [5] Saining Xie, Sainan Liu, Zeyu Chen, and Zhuowen Tu. Attentional shapecontextnet for point cloud recognition. In Proceedings of the IEEE Conference on Computer Vision and Pattern Recognition, pp. 4606–4615, 2018.

---

> ### Author Response · Authors · 2022-11-17
> **About the parameter sensitivity experiment. Thank you!**
>
> Dear Reviewer bVKL,
>
>
>
> Thank you for your suggestions to improve the quality of our paper! We have conducted parameter sensitivity experiments on the temperature and radius in contextual shape prediction loss. Results are shown as below.
>
>
>
> In the experiment on parameter sensitivity experiment on radius in contextual shape prediction, we use different radius and keep all the other parameters the same. We pre-train the 3D encoder on the DAIR-V2X dataset and downstream it to Once dataset with CenterPoint as detector. The results are shown in Table 5. It can be found that for the same R2, an increasing R1 brings more improvement on the downstream detection tasks. This might stem from that a larger inner radius can help capture more neighborhood information, making the contextual shape prediction goal more meaningful. It can be found that when R2=3.5m and R1=1m, the performance drops to 57.31. This might stem from the relatively small R2 which might fail to capture the local shape distribution of a larger object like cars or trucks. Also, we did not search these parameter before and we surprisingly find that using R1=1.5m and R2=4m brings the best performance (58.86 mAP).
>
> |        | R1(/m) |  0.5  |   1   |  1.5  |   2   |
> | :----: | :----: | :---: | :---: | :---: | :---: |
> | R2(/m) |        |       |       |       |       |
> |  3.5   |        | 57.55 | 57.31 | 58.00 | 58.42 |
> |   4    |        | 57.53 | 58.50 | 58.86 | 58.48 |
>
> **Table 5**: Results on parameter sensitivity experiments on temperature parameter.
>
>
>
> In the experiment on temperature, all other parameters are kept the same as those in the main experiment and we only change the temperature in the cooperative contrastive loss. We pre-train the 3D backbone on DAIR-V2X dataset with different parameter settings and fine-tune it on Once dataset with CenterPoint. The results are shown in Table 6. It can be found that as the temperature increases to 0.15, the downstream performance drop to 57.34 mAP. This is because a higher temperature brings smaller gap between negative pairs and this makes the representations hard to separate different objects. Also, as we keep decreasing the temperature, the overall downstream performance drop to 57.74 mAP. This is because too small temperature push representation of different objects too far away but ignore the similar semantic meaning for the same category. For example, it will make representation of two different cars far away and this will harm the performance of downstream detection task. It can also be found that using adequate temperature brings comparable performance (58.50 mAP vs 58.10 mAP).
>
> | Temperature | 0.02  | 0.07  | 0.1   | 0.15  |
> | ----------- | ----- | ----- | ----- | ----- |
> | Overall mAP | 57.74 | 58.50 | 58.10 | 57.34 |
>
> **Table 6**: Results on parameter sensitivity experiments on temperature parameter.
>
>
>
> We also add these results to Appendix H.4 in the revision. Thank you again for appreciating our work and your thoughtful suggestions!
>
>
>
> Best regards,
>
> Authors of Paper1639

---

> ### Comment · Reviewer_bVKL · 2022-11-18
> **Reply to author's rebuttal**
>
> I would like to thank the authors for addressing my comments. I believe the new experiments address my concerns on sensitivity to hyperparameters and on the magnitude of improvement (which, as mentioned in my original review, while small is consistent across methods and datasets)

---

> > ### Author Response · Authors · 2022-11-19
> > **Thank you for your reply!**
> >
> > Dear Reviewer bVKL,
> >
> > Many thanks for your reply! We are happy that our response and new experiments address your concern.
> >
> > Thank you again for your precious time on the review and the appreciation on our work.
> >
> > Best regards,
> >
> > Authors of Paper1639

---

### Official Review · Reviewer_1HxB · 2022-10-24

**Confidence:** 4
**Correctness:** 2
**Technical Novelty And Significance:** 2
**Empirical Novelty And Significance:** Not applicable
**Recommendation:** 3

**Clarity, Quality, Novelty And Reproducibility:**

The work proposes to leverage a vehicle-infrastructure-cooperation dataset that mitigates challenges due to the moving objects that would make it hard to find correct correspondence for contrastive learning. However, two major concerns are found. First, the authors did not discuss recent work by Hu et al., CVPR 2021, that aims to learn 3D point cloud representation directly from point cloud data with moving objects by formulating a freespace forecasting task. Second, the novelty of the proposed cooperative contrastive objective and contextual shape completion is questionable.

**Strength And Weaknesses:**

Strength

1. As we know, it is non-trivial to annotate point cloud data in scale. Unsupervised 3D representation learning emerges. The paper proposes an interesting approach that leverages point cloud data collected in a V2X setting. The setting mitigates challenges due to the moving objects that would make it hard to find correct correspondence for contrastive learning.
2. Empirically, the representation learned by the proposed **Co**operative **Co**ntrastive Learning and **Co**ntextual Shape Prediction (CO3) can be transferred to different outdoor scene datasets collected by different LiDAR sensors. Specifically, they show favorable performance compared with STRL (Huang et al., ICCV 2021) on the KITTI and nuScene datasets on LiDR 3D detection and semantic segmentation, respectively.

Weaknesses

1. LiDAR is indeed a valuable sensor for perception. However, it is too strong to claim that LiDAR is the most reliable sensor in outdoor environments. LiDAR has many issues, e.g., sparsity, visibility under occlusion, and false alarm due to smoke and fog. Please consider rephrasing the first paragraph.
2. The key question the authors aim to answer is: how to learn 3D representation from unlabelled point cloud data for outdoor scenes? The authors argue that existing approaches are not suitable for outdoor scenes. However, they ignore a recent advance in Freespace forecasting (Hu et al., CVPR 2021). The work could be an alternative to be compared with the proposed one. Using point clouds from both the vehicle and infrastructure sides is okay because we want data captured simultaneously. Thus, views share adequate semantics. However, the motivation is not convincing by arguing the limitations of existing self-supervised learning for point cloud data without discussing Hu et al., CVPR 2021.
- Hu et al., Safe Local Motion Planning with Self-Supervised Freespace Forecasting, CVPR 2021
3. The proposed approach's novelty is limited: This paper's two ideas are cooperative contrastive objective and contextual shape completion. The cooperative contrastive objective is the contrastive loss proposed by He et al., 2020. The only change is the input features. It seems that the contribution (cooperative contrastive objective) is over-claimed. The enabler of the proposed cooperative contrastive objective is the DAIR-V2X dataset. As mentioned by the authors, many existing approaches for shape context modeling exist. Why Eq. (5) is a better choice? Without a clear justification, we cannot conclude the importance of Eq. (5).

**Summary Of The Paper:**

The authors propose CO3, Cooperative Contrastive Learning, and Contextual Shape Prediction for unsupervised 3D representation learning in outdoor scenes. By using DAIR-V2X, a vehicle-infrastructure-cooperation dataset, the data mitigates challenges due to the moving objects that would make it hard to find correct correspondence for contrastive learning. The experiments demonstrate that the representation learned by CO3 can be transferred to various architectures and different downstream datasets and tasks to achieve performance gain.

**Summary Of The Review:**

The reviewer thinks it is interesting to leverage DAIR-V2X for unsupervised 3D representation learning. However, significant concerns are found and discussed in the Weaknesses section. The reviewer would like to know the feedback on the questions.

---

> ### Author Response · Authors · 2022-11-09
> **Response to Reviewer 1HxB**
>
> Dear Reviewer 1HxB,
>
> Thank you for your suggestions to improve the quality of our paper. We change the expression about the LiDAR sensor, add discussion on Freespace forecasting in the related work (also here), conduct new experiments on Freespace forecasting pre-training and provide an in-depth discussion about our insight and motivation in [General Response Part 2](https://openreview.net/forum?id=QUaDoIdgo0&noteId=QhrWAYDQJ3O). Also, we thank you for the suggestions on the presenation of contextual shape prediction loss. We revise the discussion in this part in Section 3.3 in the revision and also have a detailed discussion here.
>
> **Q1:** "LiDAR is indeed a valuable sensor for perception. However, ... Please consider rephrasing the first paragraph."
>
> **A1:** Thank you for pointing this out. Yes, there are several cases where LiDAR suffers such as snowing or smogging. We have changed the expression to "LiDAR is an important sensor for autonomous driving" as highlighted in blue in the first line of introduction in the revision.
>
> **Q2:** "The key question the authors aim to answer is: how to learn 3D representation from unlabelled point cloud data for outdoor scenes? The authors argue that ... However, the motivation is not convincing by arguing the limitations of existing self-supervised learning for point cloud data without discussing Hu et al., CVPR 2021."
>
> **A2:** Thank you for the suggestion! We have added the discussion on freespace forecasting in the related work section in the revision, which is highlighted in blue. Authors in [1] propose to use safe space prediction as a pre-training goal and achieve great performance in the downstream motion planning task. Compared to motion planning where understanding occupancy and forecasting safe space is more important, semantic information plays a more important role in 3D perception task and methods in [1] might not help learn representation with semantic information. As experiments about 3D perception tasks are not conducted in [1], we further conduct new experiments to pre-train 3D encoders with the pre-training goal in [1] and the results are shown in Table 4. To make fair comparison, pre-training is conducted on DAIR-V2X dataset and the downstream task is 3D object detection on Once with CenterPoint as detector. It can be found that pre-training with freespace forecasting  brings minor performance gain. We think this might stems from the objective of the pre-training goal. The pre-training goal of [1] is to predict freespace for safe driving. However, 3D perception task requires semantic information and simply predicting freespace does not help to distinguish the representation of different objects.
>
> **Table 4**: Results on using freespace forecasting as the pre-training goal. We pre-train the backbones on DAIR-V2X dataset and the downstream task is 3D object detection on Once dataset with CenterPoint.
>
> | Init                  | Vehicle | Pedestrian | Cyclist | Overall |
> | --------------------- | ------- | ---------- | ------- | ------- |
> | From Scratch          | 62.85   | 45.52      | 59.39   | 55.92   |
> | Freespace forecasting | 63.33   | 46.19      | 59.01   | 56.18   |
> | $CO3$                 | 64.60   | 48.83      | 62.17   | 58.50   |
>
>
> We also add these results to Appendix H in the revision.

---

> > ### Author Response · Authors · 2022-11-09
> > **Response of Question 3. Thank you!**
> >
> > **Q3:** "The proposed approach's novelty is limited: This paper's two ideas are cooperative contrastive objective and contextual shape completion. The cooperative contrastive objective ... Without a clear justification, we cannot conclude the importance of Eq. (5)."
> >
> > **A3:** Thank you for your advice to polish the delivery on contextual shape prediction. We have revised the expression in Section 3.3 in the revision. We also provide a detailed discussion of our motivation and insight in the [General Response Part 2](https://openreview.net/forum?id=QUaDoIdgo0&noteId=QhrWAYDQJ3O). Below, we have a brief summary and we hope our explanation can alleviate your concern.
> >
> > **Our insight is sufficient.** View building is important in contrastive learning and previous methods fail to construct suitable view for 3D unsupervised representation learning in outdoor scene. We are the first one to discover that cooperation dataset is a good source for contrastive learning. Besides, we propose the contextual shape prediction loss to introduce more task-relevant information. Also, the incooperation of the DAIR-V2X dataset for unsupervised 3D representation learning is non-trivial.
> >
> > **Contextual shape objective in equation (5).** We would like to make our claim clearer for this part and thank you for your help in improving the presentation quality of our paper. First of all, as a point feature descriptor, shape context divides the local area of the query point into several bins (8 bins in the example in Figure 4) and compute the number of points in each bins. Previous works [2,3,4,5] simply use the number of points in each bins as the point feature. Shape context is a good descriptor as demonstrated in previous work and it can be used to discribe the local neighborhood. However, it can be hard for the neural nets to directly predict the exact number of points in each bin. And we construct a local point distribution by first normalizing shape context and applying softmax to transform it to a distribution that describes the local neighborhood, which is a more suitable goal for the network to predict. New experiments using shape context in previous works as the prediction task are conducted and the results are shown in Table 1 in the [General Response Part 2](https://openreview.net/forum?id=QUaDoIdgo0&noteId=QhrWAYDQJ3O). It can be found that directly transfer the idea of shape context introduce minor performance gain.
> >
> > **CO3 will encourage the development of V2X community** We believe that CO3 will encourage future unsupervised cooperative representation learning research. Although our experiments are conducted on vehicle-infrastructure dataset, CO3 can be applied to different sources of cooperation dataset such as vehicle-to-vehicle cooperation dataset. As CO3 learns good representation without labels and V2X is a rapidly developping community, we think cooperative unsupervised representation learning on other sources can also be promising. Moreover, the original motivatio of V2X setting is to deal with the occlusion and sensor range problems. However, it is always the case that we only have sensor on the vehicle. CO3 demonstrates that pre-training with point clouds from different sources can help improve perception performance with vehicle point cloud, which is encouraging. We think CO3 will facilitate the development of the V2X community.
> >
> > Thus, we think our work is not simply applying contrastive learning and shape context on DAIR-V2X.
> >
> >
> >
> > We hope that our experiments and claims address your concern. Look forward to any further discussion.
> >
> > Best,
> >
> > Authors of paper 1639
> >
> > [1] Peiyun Hu, Aaron Huang, John Dolan, David Held, and Deva Ramanan. Safe local motion planning
> > with self-supervised freespace forecasting. In Proceedings of the IEEE/CVF Conference on
> > Computer Vision and Pattern Recognition, pp. 12732–12741, 2021.
> >
> > [2] Ji Hou, Benjamin Graham, Matthias Nie.ner, and Saining Xie. Exploring data-efficient 3d scene understanding with contrastive scene contexts. In Proceedings of the IEEE/CVF Conference on Computer Vision and Pattern Recognition, pp. 15587–15597, 2021.
> >
> > [3] Serge Belongie, Jitendra Malik, and Jan Puzicha. Shape matching and object recognition using shape contexts. IEEE transactions on pattern analysis and machine intelligence, 24(4):509–522, 2002.
> >
> > [4] Marcel K.rtgen, Gil-Joo Park, Marcin Novotni, and Reinhard Klein. 3d shape matching with 3d shape contexts. In The 7th central European seminar on computer graphics, volume 3, pp. 5–17. Citeseer, 2003.
> >
> > [5] Saining Xie, Sainan Liu, Zeyu Chen, and Zhuowen Tu. Attentional shapecontextnet for point cloud recognition. In Proceedings of the IEEE Conference on Computer Vision and Pattern Recognition, pp. 4606–4615, 2018.

---

> ### Author Response · Authors · 2022-12-05
> **Look forward to further discussion.**
>
> Dear Reviewer 1HxB,
>
> The deadline of discussion period is approaching. We hope that our response can address your concerns and we are looking forward to further discussion on any remaining question about our paper and response.
>
> Best regards,
>
> Authors of Paper 1639

---

### Official Review · Reviewer_BVa4 · 2022-10-25

**Confidence:** 5
**Correctness:** 4
**Technical Novelty And Significance:** 3
**Empirical Novelty And Significance:** Not applicable
**Recommendation:** 6

**Clarity, Quality, Novelty And Reproducibility:**

The clarity and quality of the paper are both good.
The paper also has novelty by using vehicle-infrastructure dataset for contrastive learning.
Since the experiments are based on published algorithms and datasets, this work should be reproducible.

**Strength And Weaknesses:**


Strength
1. The paper is well motivated and clearly explained.
2. The idea of using vehicle-infrastructure-cooperation dataset for unsupervised contrastive learning is interesting.
3. Experiments are extensive. Results on semantic segmentation are good.

Weakness
1. The overall improvement by the proposed CO3 is marginal for object detection. For example, the PV-RCNN results on the KITTI and ONCE datasets. If we have a larger supervised training set, will unsupervised contrastive learning still be helpful?
2. In ablation study Table 6, the results on KITTI need more explanation. What is the reason that using either of proposed contextual shape prediction and cooperative contrastive alone gives worse or marginally better performance than the random baseline but the combined method CO3 is better than the baseline?


**Summary Of The Paper:**

The paper proposes Cooperative Contrastive Learning and Contextual Shape Prediction (CO3). It features 1) using vehicle-infrastructure-cooperation dataset to obtain proper views for unsupervised contrastive learning; 2) a shape pretext for enforcing the learning to be task-relevant; and 3) the learned representation can be generalized to other autonomous driving datasets.

The proposed CO3 has been evaluated over the ONCE, KITTI and nuScenes datasets. Results show that it is effective in boosting detection performance.


**Summary Of The Review:**

I like the idea of using vehicle-infrastructure dataset for contrastive learning. But I have some concerns that the improvement on object detection is not significant.

---

> ### Author Response · Authors · 2022-11-09
> **Response to Reviewer BVa4**
>
> Dear Reviewer BVa4,
>
> Thanks for your precious time on the review and the constructive suggestions. We add repeated experiments on the KITTI dataset, provide analysis on the performance of PV-RCNN in the [General Response Part 3](https://openreview.net/forum?id=QUaDoIdgo0&noteId=Ew-KG93lBk) and add new discussion on the ablation study results.
>
> **Q1:** "The overall improvement by the proposed CO3 is marginal for object detection. For example, the PV-RCNN results on the KITTI and ONCE datasets. If we have a larger supervised training set, will unsupervised contrastive learning still be helpful?"
>
>
> **A1:**
> * **The main goal of unsupervised representation learning is to reduce labelling labor in the downstream task.** Labeling for point clouds can be very time-and-energy-consuming. Unsupervised representation learning aims to reduce the labelling labor. We have conducted experiments on large-scale supervised dataset like Waymo (150,000 frames) before and we found the performance after initialized by CO3 is only comparable to training from scratch. This phenomenon stems from the relatively small scale of the pre-training dataset. The DAIR-V2X dataset only have 6,700 frames of cooperation data. The influence of pre-training can be got rid of by finetuning on a dataset with much more data. As our method does not require labels and unlabeled cooperation dataset can be easier to collect than the labeled ones, we think a larger cooperation dataset can be collected in the future and used for unsupervised representation learning to improve downstream task with larger supervised dataset.
>
> * **The performance of PV-RCNN initialized by CO3 is remarkable on Once dataset as compared to other initialization methods.** As for the PV-RCNN performance, PV-RCNN has two branches for 3D encoding, one point-based and one voxel-based. As only the voxel-based branch is pre-trained, the improvement is relatively small as compared to CenterPoint on Once dataset (2.58 mAP). Similar phenomenon is also observed for other initialization methods. When we look at PV-RCNN on Once, it can be found that CO3 achieve the best performance and 100% more improvement than the second place (Swav). Besides, it can be found that many other initialization methods fail to improve PV-RCNN. Thus CO3 achieves remarkable improvement for PV-RCNN on Once dataset.
>
> * **The improvement on KITTI dataset is not about randomness.** As for the KITTTI dataset, the performance of different detectors are already high when training from scratch, making the performance gain seemly marginal. We conduct repeated experiments on KITTI dataset to alleviate the concerns. Results are shown in Table 2 in the [General Response Part 3](https://openreview.net/forum?id=QUaDoIdgo0&noteId=Ew-KG93lBk). First it can be found that the improvement is not about the randomness. Also,  CO3 achieves up to 300% improvement as that of strl [2] while ProposalContrast [1] even degrade the performance of Voxel-RCNN.

---

> > ### Author Response · Authors · 2022-11-09
> > **Response of question 2. We cut it into two part due to the character limitation.**
> >
> > **Q2:** "In ablation study Table 6, the results on KITTI need more explanation. What is the reason that using either of proposed contextual shape prediction and cooperative contrastive alone gives worse or marginally better performance than the random baseline but the combined method CO3 is better than the baseline?"
> >
> > **A2:** Thank you for pointing this out. We add new discussion and visualization in Appendix G in the revision to discuss this phenomenon. We also provide a discussion below for the phenomenon that using pure contextual shape prediction loss brings more improvement on Pedestrian class while cooperative contrastive loss introduce more gains on the Cyclist class.
> >
> > First of all, the "ground truth" shape context is computed with point clouds and the contextual shape prediction goal is to predict the local point distribution with voxel-level representation. This enables the voxel-level representation to predict the structure inside a voxel. Meanwhile, cooperative contrastive only focus on contrasting different voxel representation. For pedestrian class, there are usually only one or two voxels for one pedestrian. Thus, with cooperative contrastive loss only, the representation fail to recognize the inner structure of the voxel and thus bring small improvement upon the pedestrian category. On the contrary, when we pre-train the encoder with contextual shape prediction loss, the learned representation is able to express the structure inside the voxel and this is helpful in the downstream detection on Pedestrian class.
> >
> > Second, as the contextual shape prediction goal aims to capture local shape distribution, it fails to learn good representation with varying shape. Compared to pedestrians whose shape is always cylinder-like, cyclists with their bicycles are usually captured with different poses, leading to different shapes. A visualization of different cases of the two categories are provided in Figure 6 in Appendix G in the revision. Thus if we use contextual shape prediction as the only pre-training goal, the representation fails to capture semantic information of cyclist class with varying shape. In Table 6, it can be found that pre-training with contextual shape loss only cannot bring significant improvement on Cyclist class.
> >
> > Besides, as discussed in Appendix B, the two pre-training objectives are complementary. Pure cooperative contrastive learning makes the representation minimal sufficient, which lacks of task-relevant information. The contextual shape prediction loss brings more task-relevant information by increasing the mutual information between the representations and the inputs. Thus combining them leads to better performance.
> >
> >
> >
> >
> >
> > We hope that our replies and experiments address your concern and we are happy for any further discussion.
> >
> > Best,
> >
> > Authors of paper 1639
> >
> > [1] Junbo Yin, Dingfu Zhou, Liangjun Zhang, Jin Fang, Cheng-Zhong Xu, Jianbing Shen, and Wenguan Wang. Proposalcontrast: Unsupervised pre-training for lidar-based 3d object detection. 2022
> >
> > [2] Siyuan Huang, Yichen Xie, Song-Chun Zhu, and Yixin Zhu. Spatio-temporal self-supervised representation learning for 3d point clouds. In Proceedings of the IEEE/CVF International Conference on Computer Vision, pp. 6535–6545, 2021.

---

### Official Review · Reviewer_XpxU · 2022-10-26

**Confidence:** 4
**Correctness:** 4
**Technical Novelty And Significance:** 2
**Empirical Novelty And Significance:** 2
**Recommendation:** 5

**Clarity, Quality, Novelty And Reproducibility:**

The paper is clearly written. The novelty of the method itself is limited.
However, the reproducibility is high since most of the components are off-the-shelf.

**Strength And Weaknesses:**

Strength
+ The motivation of construct views from infrastructure sensors and vehicle sensors is well introduced.
+ The performance gain is significant on various downstream benchmarks.
+ The evaluation is extensive.

Weakness:
+ The paper converts the view construction of outdoor scene to a case similar to indoor by using the new dataset. So the proposed solution should also works for indoor, but which is not demonstrated in the paper.
+ The novelty and the insight of the method itself is limited.

**Summary Of The Paper:**

This paper focuses on self-supervised point cloud representation learning. The authors combine the contrastive loss with KL divergence between the predicted feature and manually crafted 3D shape context.

To construct better views for contrastive loss,  the authors utilize a recently proposed dataset where point clouds are captured from both vehicle sensors and infrastructure sensors.

To demonstrate the effectiveness of the proposed methods, the authors finetuned the pre-trained model on  3D object detection and segmentation tasks and have achieved better performance than other self-supervised learning methods or even supervised pre-trained models.

**Summary Of The Review:**

Although the performance gain is significant, I prefer to reject the paper because it resolves the (positive) view construction of 3D point clouds by using an existing dataset, and the novelty and insights of the method itself are insufficient.


[post-rebuttal] I appreciate the efforts the authors did on proving the effectiveness of the proposed method on indoor data. Although I maintain my original opinion that the paper is more about construct data on a very specific dataset and is less inspiring for further researches in general case, I updated the rating due to the additional effort the authors have made.

---

> ### Author Response · Authors · 2022-11-09
> **Response to Reviewer XpxU**
>
> Dear Reviewer XpxU,
>
> Thanks for your review. We conduct new experiments for indoor scene point clouds and provide a clarification on the motivation and insight both in [General Response Part 2](https://openreview.net/forum?id=QUaDoIdgo0&noteId=QhrWAYDQJ3O) and here.
>
> **Q1:** "The paper converts the view construction of outdoor scene to a case similar to indoor by using the new dataset. So the proposed solution should also works for indoor, but which is not demonstrated in the paper."
>
> **A1:** Thank you for your suggestions. We conduct new experiments on indoor scene point clouds. We use the ScanNet dataset [2] to pre-train the encoder with PointContrast [1] and our CO3. The pre-training on the whole ScanNet dataset can be very time-consuming and we adopt a more efficient training schedule as stated in the PointContrast [1] repository. The downstream task is object detection on the SUNRGBD dataset [3] and we choose the VoteNet [4] as the detector. The same pre-training and downstream settings are used for both initialization methods. The results are shown in Table 3. The results show that our method is also suitable for indoor scene point cloud representation learning and achieves more than 50% improvement over PointContrast [1]. Thus our insight works not only for outdoor scene point clouds but also for indoor scene point clouds.
>
> **Table 3**: Results of experiments for indoor scene point clouds. The encoder is pre-trained on ScanNet dataset and downstream to VoteNet on 3d object detection on SUNRGBD dataset.
>
> | Init          | mAP               |
> | ------------- | ----------------- |
> | Random (from scratch)  | 31.70             |
> | PointContrast | 32.65 (+0.95 mAP) |
> | CO3           | 33.10 (+1.40 mAP) |
>
>
>
>
>
> **Q2:** "The novelty and the insight of the method itself is limited."
>
> **A2:** We provide an in-depth discussion of our motivation and insight in the [General Response Part 2](https://openreview.net/forum?id=QUaDoIdgo0&noteId=QhrWAYDQJ3O) in order to alleviate your concern. Thanks for your suggestions to improve our delivery. Below we also provide a brief summary.
>
> **Our insight is sufficient.** View building is important in contrastive learning and previous methods fail to construct suitable view for 3D unsupervised representation learning in outdoor scene. We are the first one to discover that cooperation dataset is a good source for contrastive learning. Also, the incorporation of the DAIR-V2X dataset for unsupervised 3D representation learning is non-trivial. We propose to use fusion point cloud as another view for cooperative contrative learning and also a contextual shape prediction loss to introduce more task-relevant information.
>
> **CO3 will encourage the development of V2X community.** We believe that CO3 will encourage future cooperative unsupervised representation learning research. Although our experiments are conducted on vehicle-infrastructure dataset, CO3 can be applied to different sources of cooperation dataset such as vehicle-to-vehicle cooperation dataset. As CO3 learns good representation without labels and V2X is a rapidly developping community, we think cooperative unsupervised representation learning on other sources can also be promising. Moreover, the original motivation of V2X setting is to deal with the occlusion and sensor range problems. However, it is always the case that we only have sensor on the vehicle. CO3 demonstrates that pre-training with point clouds from different sources can help improve perception performance with vehicle point cloud, which is encouraging. We think CO3 will facilitate the development of the V2X community.
>
> **CO3 improves the representation learning in indoor scene.** Moreover, applying our insight to indoor scene point cloud provide more improvements than previous methods for indoor scene point clouds.
>
> Thus, we think our work is not simply applying contrastive learning on a new dataset.
>
>
> We hope that our reply address your concern and we are happy for any further discussion.
>
> Best regards,
>
> Authors of paper 1639
>
>
> [1] Saining Xie, Jiatao Gu, Demi Guo, Charles R Qi, Leonidas Guibas, and Or Litany. Pointcontrast: Unsupervised pre-training for 3d point cloud understanding. In European Conference on Computer Vision, pp. 574–591. Springer, 2020.
>
> [2] Dai, Angela and Chang, Angel X. and Savva, Manolis and Halber, Maciej and Funkhouser, Thomas and Nie{\ss}ner, Matthias. ScanNet: Richly-annotated 3D Reconstructions of Indoor Scenes. In Proc. Computer Vision and Pattern Recognition (CVPR), IEEE, 2017.
>
> [3] S. Song, S. Lichtenberg, and J. Xiao. SUN RGB-D: A RGB-D Scene Understanding Benchmark Suite. In Proc. Computer Vision and Pattern Recognition (CVPR), IEEE, 2015.
>
> [4] Qi C R, Litany O, He K, et al. Deep hough voting for 3d object detection in point clouds. In proceedings of the IEEE/CVF International Conference on Computer Vision. 2019.

---

> ### Author Response · Authors · 2022-12-05
> **Look forward to further discussion.**
>
> Dear Reviewer XpxU,
>
> We hope that our response can address your concerns. As the deadline for discussion period is approaching, we really appreciate if you can let us know whether there still exists any further question about the paper or the response. We are looking forward to further discussion.
>
> Best regards,
>
> Authors of Paper 1639

---

### Author Response · Authors · 2022-11-09
**General Response (Part 3: About the magnitude of performance improvement on PV-RCNN.)**

**General Response Part 3: About the magnitude of performance improvement on PV-RCNN.**

* **The performance improvement of PV-RCNN is remarkable on Once as compared to other initialization methods.** PV-RCNN has two 3D encoders, one point-based and one voxel-based. Because only the voxel-based branch is pre-trained, the improvement is relatively small as compared to CenterPoint on Once dataset (2.58 mAP). When we look at different initialization methods on PV-RCNN, it can be found that CO3 achieve superior performance. For example, on Once dataset, Swav achieve the second place on overall mAP for PV-RCNN. However, when we compare the performance gain, CO3 achieves 100% more performance gain than Swav. Also, it can be found that many other initialization methods fail to improve PV-RCNN.
* **The improvement on KITTI dataset is not about randomness.** The performance of different detectors on KITTI dataset is already high when training from scratch, making the performance gain seemly marginal. To alleviate the concerns about performance on KITTI dataset, we conduct repeated experiments on KITTI dataset to demonstrate the stability of CO3 in improving performance. As the training of PV-RCNN is time-consuming, we conduct experiments on a smaller detector (Voxel-RCNN [1]) on KITTI dataset for 5 times and compute the mean and variance of the results. In Table 2, it can be found that the improvement is not about the randomness. Also,  CO^3 achieves more than 300% improvement as that of strl [2] while ProposalContrast [3] even degrade the performance.

**Table 2**: Results of repeated experiments for Voxel-RCNN [1] trained simply on vehicle class on KITTI dataset. We repeat the downstream training for 5 times (different random seeds) for each initialization method and compute the mean and variance of these 5 experiments. Results are shown in three difficult level: Easy, Moderate and Hard.

| Init         | Easy             | Moderate         | Hard             |
| ------------ | ---------------- | ---------------- | ---------------- |
| From Scratch | $89.12\pm0.0030$ | $79.00\pm0.0020$ | $78.24\pm0.0008$ |
| ProposalContrast        | $88.95\pm0.0020$ | $78.95\pm0.0032$ | $78.24\pm0.0010$ |
| strl         | $89.20\pm0.0033$ | $79.05\pm0.0022$ | $78.31\pm0.0005$ |
| $CO^3$       | $89.46\pm0.0021$ | $79.27\pm0.0033$ | $78.44\pm0.0007$ |

[1] Deng, Jiajun and Shi, Shaoshuai and Li, Peiwei and Zhou, Wengang and Zhang, Yanyong and Li, Houqiang. Voxel R-CNN: Towards High Performance Voxel-based 3D Object Detection. arXiv:2012.15712. 2020.

[2] Siyuan Huang, Yichen Xie, Song-Chun Zhu, and Yixin Zhu. Spatio-temporal self-supervised representation learning for 3d point clouds. In Proceedings of the IEEE/CVF International Conference on Computer Vision, pp. 6535–6545, 2021.

[3] Junbo Yin, Dingfu Zhou, Liangjun Zhang, Jin Fang, Cheng-Zhong Xu, Jianbing Shen, and Wenguan Wang. Proposalcontrast: Unsupervised pre-training for lidar-based 3d object detection. 2022

---

### Author Response · Authors · 2022-11-09
**General Response (Part 2: About novelty and insight of our method.)**

**General Response Part 2: About novelty and insight of our method.**

We thank reviewer XpxU and 1HxB for their suggestions. We polish our delivery of the insight and novelty of CO3 and provide an in-depth discussion here. We will first discuss our insight. Then, we talk about the influence of CO3 on the V2X community. Finally, we have a discussion on using cooperative dataset for unsupervised representation learning.

* **Our insight is sufficient.**
    * **Cooperative views makes good views for contrastive learning in autonomous driving.** Previous work on unsupervised representation learning for outdoor scene point cloud build inadequate views for contrastive learning. As shown in Figure 1 (b) in our paper, [1,2] use linear transformation to augment single frame of pointcloud as views in contrastive learning. However, as the non-linear transformation in [3] already brings similiar views, views built with linear augmentation differ too little. [4] proposes to use point clouds at different timestamps but this will make it hard to find correct correspondence due to the dynamic property of ourdoor scenes. Thus in this paper, we propose to use cooperative views for unsupervised representation learning.
    * **Contextual shape prediction brings task-relevant information.** Pure contrastive learning makes the representation lack of task-relevant information and a reconstruction goal can compensate this, which is discussed and proved in Appendix B. However, for large-scale outdoor scene, it can be extremely hard for voxel/point-level representations to recover the whole scene. So we propose to reconstruct local neighborhood distribution, which is described by an extension of shape context descriptor.

* **Our work encourages future attempts in cooperative pre-training.** V2X community is developing rapidly [5] and in the future there will be more relevant datasets in V2X setting. Although our experiments are done on the vehicle-infrastructure-cooperation dataset, our methods can also be applied to other V2X setting like vehicle-to-vehicle cooperation dataset. We believe that the promising performance of learning general representation would encourage future efforts on other cooperative settings. Also, the most time-and-energy consuming part of collecting cooperation dataset is labeling. As CO3 does not need any label, the collection of datasets can be much easier and more scalable. We think larger unlabeled cooperative datasets will be collected in the future and used for unsupervised 3D representation learning to achieve more performance gain.

* **Our work will promote the development of the V2X community.** The original motivation for V2X community is about incooperating more information from different sources to deal with the long-range sensing and occlusion problem. However, we only have sensor on the vehicle in many areas and it would be too expensive to set sensors everywhere. In our paper, V2X setting helps view building in unsupervised representation learning and improve 3D perception tasks using point clouds from vehicle side. This is encouraging and we believe the promising results on the learned representation will promote the development of the V2X community.

---

> ### Author Response · Authors · 2022-11-09
> **It is non-trivial to incorporate the cooperation dataset for unsupervised 3D representation learning.**
>
> * **It is non-trivial to incorporate the cooperation dataset for unsupervised 3D representation learning.**
>     * **Directly apply contrastive learning on point clouds from different sources does not work.** Building views directly from the vehicle and infrastructure-side shows little performance improvement as shown in Table 7 in appendix D. We think this might stem from the sparsity of the LiDAR point cloud. Thus we propose to use the fusion point clouds in contrastive learning. Also, as shown in Table 6 of our ablation study, simply use cooperative contrastive learning makes the improvement drops by 40% and 80% on the Once and KITTI dataset respectively as compared to our CO3. We provide an analysis on this phenomenon in Appendix B and propose to use the contextual shape prediction loss to introduce more task-relevant information.
>     * **Contextual shape prediction loss formulates a better pre-training goal than previous work.** We would like to make our claim clearer for this part. First of all, as a point feature descriptor, shape context divides the local area of the query point into several bins (8 bins in the example in Figure 4) and compute the number of points in each bins. Previous works [6,7,8,9] simply use the number of points in each bins as the point feature. Shape context is a good descriptor as demonstrated in previous work and it can be used to discribe the local neighborhood. However, it can be hard for the neural nets to directly predict the exact number of points in each bin. To make the prediction goal more suitable for the neural net to learn, we construct a local point distribution by first normalizing the shape context and applying softmax to transform the shape context descriptor to a distribution that describes the local neighborhood. This finally arrives in our proposed CO3. Here we also conduct a new ablation experiment where the reconstruction goal is to predict the exact number of points in each bin. The results are shown in Table 1 and it can be found that directly transfer the idea of shape context only brings much lower improvement (0.94mAP) as compared to the proposed CO3 (2.58 mAP).
>
>
> **Table 1**: Results on using shape context in previous work for pre-training.
>
> | Init                                                         | Vehicle | Pedestrian | Cyclist | Overall |
> | ------------------------------------------------------------ | ------- | ---------- | ------- | ------- |
> | From Scratch                                                 | 62.85   | 45.52      | 59.39   | 55.92   |
> | $CO3$ with predicting exact number of the points in shape context | 62.79   | 47.63      | 60.15   | 56.86   |
> | $CO3$                                                        | 64.60   | 48.83      | 62.17   | 58.50   |
>
> Thus we think our proposed CO3 is not simply using new dataset for contrastive learning. Also we provide new explanation on the contextual shape prediction loss and hope that can address the confusion and make our presentation clearer. We will add the discussion about influence on V2X community in the final version.
>
> [1] Hanxue Liang, Chenhan Jiang, Dapeng Feng, Xin Chen, Hang Xu, Xiaodan Liang, Wei Zhang, Zhenguo Li, and Luc Van Gool. Exploring geometry-aware contrast and clustering harmonization for self-supervised 3d object detection. ICCV 2021.
>
> [2] Junbo Yin, Dingfu Zhou, Liangjun Zhang, Jin Fang, Cheng-Zhong Xu, Jianbing Shen, and Wenguan Wang. Proposalcontrast: Unsupervised pre-training for lidar-based 3d object detection. 2022
>
> [3] Yonglong Tian, Chen Sun, Ben Poole, Dilip Krishnan, Cordelia Schmid, and Phillip Isola. What makes for good views for contrastive learning? In H. Larochelle, M. Ranzato, R. Hadsell, M. F. Balcan, and H. Lin NeurIPS 2020.
>
> [4] Siyuan Huang, Yichen Xie, Song-Chun Zhu, and Yixin Zhu. Spatio-temporal self-supervised representation learning for 3d point clouds. ICCV 2021.
>
> [5] Haibao Yu, Yizhen Luo, Mao Shu, Yiyi Huo, Zebang Yang, Yifeng Shi, Zhenglong Guo, Hanyu Li, Xing Hu, Jirui Yuan, and Zaiqing Nie. Dair-v2x: A large-scale dataset for vehicle-infrastructure cooperative 3d object detection.CVPR 2022.
>
> [6] Ji Hou, Benjamin Graham, Matthias Nie.ner, and Saining Xie. Exploring data-efficient 3d scene understanding with contrastive scene contexts. CVPR 2021.
>
> [7] Serge Belongie, Jitendra Malik, and Jan Puzicha. Shape matching and object recognition using shape contexts. IEEE transactions on pattern analysis and machine intelligence, 2002.
>
> [8] Marcel K.rtgen, Gil-Joo Park, Marcin Novotni, and Reinhard Klein. 3d shape matching with 3d shape contexts. In The 7th central European seminar on computer graphics, volume 3, pp. 5–17. Citeseer, 2003.
>
> [9] Saining Xie, Sainan Liu, Zeyu Chen, and Zhuowen Tu. Attentional shapecontextnet for point cloud recognition.CVPR 2018.

---

### Author Response · Authors · 2022-11-09
**General Response (Part 1: summary of the reviews and rebuttal)**

Dear AC and reviewers,



Thank you for your precious time on the review and your constructive suggestions to improve our manuscript! We appreciate that reviewers acknowledge our idea is interesting and well-motivated (Reviewer BVa4, Reviewer 1HxB, Reviewer bVKL), the experiments are extensive (Reviewer XpxU, Reviewer BVa4, Reviewer 1HxB, Reviewer bVKL) and our presentation is clear and easy-to-follow (Reviewer XpxU, Reviewer BVa4, Reviewer bVKL).



Here are the summary about our new experiments in the rebuttal and the revision of the paper.

**New Experiments**

* Experiments for indoor scene point clouds. (Reviewer XpxU)
* Experiments using the safespace prediction in [1] to pre-train the 3D encoder. (Reviewer 1HxB)
* Ablation study of using shape context in previous works [2,3,4,5] as the pre-training goal.
* Repeated experiments on KITTI datasets.
* Parameter sensititive experiments, which is still ongoing and we will update the results as soon as they are finished. (Reviewer bVKL)

**Revisions (highlighted in blue in the revised paper)**

* We add discussion of safespace prediction [1], task-relevant loss and shape context in the related work.
* We revise our discussion on local distribution building in Section 3.3 to avoid confusing readers about our contextual shape prediction loss.
* We add more detailed discussion for the ablation study in Appendix G.
* We add the results of new experiments in Appendix H.
* We add expressions on the subscript 'v/f' before and after equation (1) to avoid confusion.
* We revise some strong claims in the original version and add the results in Table 7 to the text part in the main paper.


We will first address common questions in the General Response [Part 2](https://openreview.net/forum?id=QUaDoIdgo0&noteId=QhrWAYDQJ3O) and [Part 3](https://openreview.net/forum?id=QUaDoIdgo0&noteId=Ew-KG93lBk) and answer separate concerns from each reviewer. The common concerns come from (1) the delivery of novelty and insight of our method (Reviewer XpxU and Reviewer 1HxB) ([Part 2](https://openreview.net/forum?id=QUaDoIdgo0&noteId=QhrWAYDQJ3O))  and (2) the magnitude of performance improvement on PV-RCNN (Reviewer BVa4, Reviewer bVKL) ([Part 3](https://openreview.net/forum?id=QUaDoIdgo0&noteId=Ew-KG93lBk)).

We hope that our reply, new experiments and revision address your concern. Look forward to further discussion!

Best regards,
Authors of paper 1639

[1] Peiyun Hu, Aaron Huang, John Dolan, David Held, and Deva Ramanan. Safe Local Motion Planning with Self-Supervised Freespace Forecasting, CVPR 2021

[2] Ji Hou, Benjamin Graham, Matthias Nie.ner, and Saining Xie. Exploring data-efficient 3d scene understanding with contrastive scene contexts. In Proceedings of the IEEE/CVF Conference on Computer Vision and Pattern Recognition, pp. 15587–15597, 2021.

[3] Serge Belongie, Jitendra Malik, and Jan Puzicha. Shape matching and object recognition using shape contexts. IEEE transactions on pattern analysis and machine intelligence, 24(4):509–522, 2002.

[4] Marcel K.rtgen, Gil-Joo Park, Marcin Novotni, and Reinhard Klein. 3d shape matching with 3d shape contexts. In The 7th central European seminar on computer graphics, volume 3, pp. 5–17. Citeseer, 2003.

[5] Saining Xie, Sainan Liu, Zeyu Chen, and Zhuowen Tu. Attentional shapecontextnet for point cloud recognition. In Proceedings of the IEEE Conference on Computer Vision and Pattern Recognition, pp. 4606–4615, 2018.

---

### Author Response · Authors · 2022-11-18
**Summary of rebuttal phase.**

Dear AC and reviewers,



Many thanks for your valuable comments and suggestions to improve the quality of our work. Here is a summary of what we have done in the rebuttal phase.



* We made several revision to our original manuscript as stated in the [Paper update part](https://openreview.net/forum?id=QUaDoIdgo0&noteId=6wCzw8aWS4).
* We conducted several new experiments to cover the concerns from the reviewers.
  * Applying our proposed method to indoor scene point clouds.
  * Comparison to safety space forecasting pre-training [1].
  * Parameter sensitivity experiments on temperature and radius in the contextual shape prediction loss.
  * Ablation study on using the original shape context descriptor as the prediction goal.
  * Repeated experiments on KITTI dataset to demonstrate the stability of the improvement.
* We provide an in-depth discussion about our insight and novelty. We also discuss the influence of our work on the V2X community. Details can be found in [General Response Part 2](https://openreview.net/forum?id=QUaDoIdgo0&noteId=QhrWAYDQJ3O).
* We provide a detailed discussion on the improvement magnitude on PVRCNN on KITTI dataset in [General Response Part 3](https://openreview.net/forum?id=QUaDoIdgo0&noteId=Ew-KG93lBk).
* We provide a discussion about the results in ablation study in Appendix G and also the response to [Reviewer Bva4](https://openreview.net/forum?id=QUaDoIdgo0&noteId=2ITGHc14EBN).
* We provide a discussion on how the cooperation dataset is collected and why it is scalable to collect such dataset without labels in the response to [Reviewer bVKL](https://openreview.net/forum?id=QUaDoIdgo0&noteId=k08uOU-L6T).



Thank you again for your precious time on the review. We hope that our response has addressed your concerns. We are happy to have further discussion on anything unclear about our paper.



Best regards,

Authors of Paper1639

[1] Peiyun Hu, Aaron Huang, John Dolan, David Held, and Deva Ramanan. Safe Local Motion Planning with Self-Supervised Freespace Forecasting, CVPR 2021

---

### Author Response · Authors · 2022-11-18
**Paper update.**

Dear AC and reviewers,



Thank you for your precious time and constructive suggestions to improve the quality of our paper. In the rebuttal period, we made several update to our manuscript, which are highlighted in blue in the revision, and we provide a summary below:

* We add discussion of safespace prediction [1], task-relevant loss and shape context in the related work.
* We revise our discussion on local distribution building in Section 3.3 to avoid confusing readers about our contextual shape prediction loss.
* We add more detailed discussion for the ablation study in Appendix G.
* We discuss the influence of our work on the V2X community in Appendix I to emphasize the novelty and significance in V2X community.
* We add the results of new experiments, including indoor scene experiments, comparison to safety space forecasting pre-training and parameter sensitivity experiements in Appendix H.
* We add expressions on the subscript 'v/f' before and after equation (1) to avoid confusion.
* We revise some strong claims in the original version and add the results in Table 7 to the text part in the main paper.



We hope that our revision makes our presentation clearer and cover reviewers' concern. We are happy to answer any further question about our work.

Best regards,

Authors of Paper1639



[1] Peiyun Hu, Aaron Huang, John Dolan, David Held, and Deva Ramanan. Safe Local Motion Planning with Self-Supervised Freespace Forecasting, CVPR 2021

---

### Decision · Program_Chairs · 2023-01-20

**Decision:**

Accept: poster

**Justification For Why Not Higher Score:**

There remain concerns regarding the novelty of the formulation, as well as the general applicability of the work.

**Justification For Why Not Lower Score:**

Most reviewers appreciate the extensive experiments and the good results.

**Metareview: Summary, Strengths And Weaknesses:**

This paper studies unsupervised representation learning on 3D scenes. Reviewers like the motivation and the strong results; however, initially there are concerns about the experiments being very specific and the formulation not being novel enough. The authors did a great job during the rebuttal and reviewers are in general more positive afterward.  Reviewer 1HxB indicated that their concerns were mostly addressed with the new experiments and ablations and would like to raise the score to accept (though they didn't update the form).  Reviewer XpxU also updated the rating based on the new results on indoor scenes.  Based on all feedback, the AC agrees that this paper can be accepted. The authors are encouraged to revise the paper to incorporate these comments and the new results.

**Note From Pc:**

if the above contains the word "oral" or "spotlight" please see: "oral" presentation means -> notable-top-5% and "spotlight" means -> notable-top-25%. As stated in our emails, we are disassociating presentation type from AC recommendations